# Introducing "Forecast Utterance" for Conversational Data Science

**Md Mahadi Hassan**[*]                                                                    *mzh0167@auburn.edu*
*Department of Computer Science and Software Engineering*
*Auburn University*

**Alex Knipper**[*]                                                                        *rak0035@auburn.edu*
*Department of Computer Science and Software Engineering*
*Auburn University*

**Shubhra Kanti Karmaker (Santu)**                                                         *sks0086@auburn.edu*
*Department of Computer Science and Software Engineering*
*Auburn University*

**Reviewed on OpenReview:** *https://openreview.net/forum?id=H2EeStRTQn*

## Abstract

Envision an intelligent agent capable of assisting users in conducting forecasting tasks through intuitive, natural conversations without requiring in-depth knowledge of the underlying machine learning (ML) processes. A significant challenge for the agent in this endeavor is to accurately comprehend the user's prediction goals and, consequently, formulate precise ML tasks. In this paper, we take a pioneering step towards this ambitious goal by introducing a new concept called *Forecast Utterance* and then focus on the automatic and accurate interpretation of users' prediction goals from these utterances. Specifically, we frame the task as a slot-filling problem, where each slot corresponds to a specific aspect of the goal prediction task. We then employ two methods based on self-supervision with synthetic examples for solving the slot-filling task, namely: 1) Entity Extraction (EE), and 2) Question-Answering (QA) techniques. Our experiments, evaluated with three meticulously crafted data sets, validate the viability of our ambitious goal and demonstrate the effectiveness of both EE and QA techniques in interpreting *Forecast Utterances*.

## 1 Introduction

Imagine a conversational AI agent designed to assist end-users, who are not experts in machine learning, with simple forecasting tasks such as estimating a publicly traded stock's future price or predicting the average temperature of a geographic location in the upcoming week. One critical challenge for such an agent is to accurately understand the user's forecasting goals and formulate a precise machine learning task accordingly. As these conversations are expected to happen in real-time, and each user may have unique data sets and data science needs (Parsons et al., 2011), it is unrealistic to assume that any training data is available to pre-train these conversational agents, making the supervised learning paradigm impractical. As such, the central question we investigate in this paper is how to automatically and accurately understand users' forecasting goals from their utterances in an unsupervised fashion.

**Motivation:** Time series forecasting is an essential tool for informed decision-making and strategic planning across various organizations. It offers valuable insights into future business developments, including revenue, sales, and resource demand. However, small businesses and machine learning enthusiasts from diverse backgrounds often face challenges in harnessing the benefits of time series forecasting due to several factors: 1) Limited ML-related technical expertise among entrepreneurs and business owners; 2) A lack of highly competent data science teams in small businesses; and 3) The high cost of hiring external consultants and data privacy concerns when involving third

---

[*]Md Mahadi Hassan and Alex Knipper contributed equally to this work.

parties. In this context, a conversational approach that simplifies the application of time series forecasting would be highly appealing. Furthermore, time series forecasting is primarily self-supervised in nature as it relies mostly on historical data without requiring additional human labels, making it an ideal candidate for formulating ML tasks in a self-supervised fashion. With the growing popularity of AI chatbots, the proposed "Conversational Data Science" approach can significantly broaden ML's reach and impact across industries.

**The Conceptual Leap:** Existing *ML* solutions demand a clear understanding of ML techniques on the user's end as well as requires significant manual effort for executing the end-to-end Data Science Pipeline, making Data Science inaccessible to the general public Karmaker et al. (2021). However, Conversational AI research holds the potential to democratize data science for a broader audience. This paper connects machine learning pipeline automation with Conversational AI, creating a novel solution for forecasting task formulation by integrating simple yet powerful methods from both disciplines.

**Challenges:** In real-time conversations, end users will provide their dataset "on the fly" and try to formulate a forecasting task on the provided dataset. This makes it impossible to pre-train an ML task formulation model, as the attribute set is different each time. To address this challenge, we frame it as an unsupervised slot-filling problem, where each aspect of the prediction goal represents a slot. We then implement two approaches that leverage self-supervision with synthetic examples to populate the slots: 1) Entity Extraction (EE) and 2) Question-Answering (QA) techniques.

**Contributions:** This paper takes an initial step towards formulating prediction tasks through natural conversation by focusing on a specific type of user utterance called a ***Forecast Utterance***. We design and develop techniques for automated understanding of users' forecasting goals from these utterances in an unsupervised fashion. The main contributions of this paper include the following:

• Introducing the *Forecast Utterance*, an expression of users' prediction needs and goals via natural language, and creating three benchmark datasets, through extensive manual effort, to promote future research in this area.
• Framing *Forecast Utterance Understanding* as an unsupervised slot-filling problem, where every prediction need is represented as a distinct slot. To address this task of slot-filling, our approach involves the application of Entity Extraction (EE) and Question-Answering (QA) strategies. Both strategies utilize self-supervision by generating synthetic examples.
• Conducting case studies with three real-world datasets to demonstrate the feasibility and efficacy of the proposed ideas and techniques.

## 2 Related Works

Over the past decade, the machine learning community has made significant advancements in automating machine learning pipelines. Despite these developments, automating Prediction Task Formulation remains a challenge due to its human-centric nature (Karmaker et al., 2021; Sarkar et al., 2023). In parallel, NLP researchers have made significant progress in domains such as Dialog Systems (Weizenbaum, 1966), Slot Filling (Lafferty et al., 2001a), Zero-Shot Learning (Palatucci et al., 2009; Sarkar et al., 2022), E-commerce search (Santu et al., 2017) and many more, to create more sophisticated conversational agents. For example, Dialog Systems research has evolved through Conversation Topic Prediction, Dialogue State Tracking, and open-domain dialogue system innovations, with Large Language Models (LLMs) like ChatGPT [1] being among the latest developments. On the other hand, the Slot Filling problem has been addressed as a sequence labeling task by leveraging CRFs (Lafferty et al., 2001b), RNNs (Williams and Zipser, 1989), and self-attention transformers (Vaswani et al., 2017), while Self-Supervised Learning (Akata et al., 2013) has primarily focused on recognizing unseen labels and has become a very popular paradigm in ML research recently. In this section, we will delve deeper into these areas and discuss their relevance to our research.

**Dialog Systems:** In Dialog Systems research, significant progress has been achieved through advancements in Conversation Topic Prediction (Khatri et al., 2018) and Dialogue State Tracking (DST) (Henderson et al., 2014a;b). DST improvements involve a range of approaches, including schema guidance for better structure (Chen et al., 2020; Zhu et al., 2020; Kapelonis et al., 2022), recursive inference for deeper understanding (Liao et al., 2021), generalization and value normalization for more adaptability (Williams, 2013; Wang et al., 2020), zero-shot transfer learning for

---

[1] https://openai.com/blog/chatgpt

data efficiency (Campagna et al., 2020; Rabinovich et al., 2022), and attention modulation for improved focus during inference (Veron et al., 2022). Open-domain dialogue systems have also seen significant advancements. GODEL's (Peng et al., 2022) grounded pre-training adapts to diverse downstream tasks, FusedChat (Young et al., 2022) combines task-oriented and open-domain dialogue for natural conversations, & ChatGPT further enhances conversational agent performance across various applications such as data science (Hassan et al., 2023).

**Slot Filling:** Slot Filling has been studied across applications like recommender systems and chatbots, using approaches such as RNNs (Kurata et al., 2016; Mesnil et al., 2015), integrating CRFs with RNNs (Huang et al., 2015; Reimers and Gurevych, 2017; Jbene et al., 2022), and self-attention mechanisms for sequence labeling (Shen et al., 2018; Tan et al., 2018; Zhao et al., 2022). Joint learning of intent detection and slot-filling has been explored (Liu and Lane, 2016; Goo et al., 2018; Zhang et al., 2019; Chen and Luo, 2023), incorporating few-shot learning and model-agnostic meta-learning (Bhathiya and Thayasivam, 2020; Krone et al., 2020). Transfer learning has led to zero-shot slot filling approaches (Mehri and Eskenazi, 2021; Wang et al., 2021; Larson and Leach, 2022), enhancing knowledge transfer between pre-trained models and target domains, improving performance on unseen slots and achieving state-of-the-art results.

**Self-Supervised Learning with Synthetic Examples:** Recent self-supervised learning research explores synthetic data to improve model robustness and adaptability. Leveraging synthetic datasets enhances contrastive learning, leading to more generalized feature representations (She and Xu, 2021). Additionally, the critical role of advanced data augmentation in disentangling content from style and boosting out-of-domain robustness has been emphasized (von Kügelgen et al., 2021; Ng et al., 2020). Notably, the potential of synthetic data to empower neural machine translation, particularly for resource-scarce languages, highlights its growing influence (Ruiter et al., 2021). These advancements solidify the pivotal role of synthetic and augmented data in propelling the frontiers of self-supervised learning. Again, recent research showcases — time series forecasting using meta-learning frameworks (Oreshkin et al., 2021; Abdallah et al., 2022; Van Ness et al., 2023), feature construction for time series variables(Wang et al., 2017), layer ensemble architecture(Rahman et al., 2014), data augmentation & adversarial domain adaptation (Hu et al., 2020; Jin et al., 2022), and ordinal regression recurrent neural networks (Orozco and Roberts, 2020). Also researcher has tried to recommend prediciton task to users interactively (Xu et al., 2019). However, these approaches don't address Forecast Utterance Understanding for unseen datasets in real-time, which is this paper's primary focus.

**Difference from Previous Work:** Our work distinguishes itself from previous works by introducing a novel concept called "Forecast Utterance" and demonstrating the feasibility of so-called "Conversational Data Science". In contrast to existing Dialog Systems and Slot-Filling research, our primary focus is on understanding an end-user's forecasting needs by framing it as a slot-filling task where each slot represents a unique aspect of the goal prediction task. Additionally, we propose a novel synthetic data generation technique to fine-tune pre-trained transformer-based models for Entity Extraction (EE) and Question Answering (QA) tasks to perform inference in real-time.

## 3 Problem Definition

We treat the task as a slot-filling problem, with each aspect of the user's prediction need as a slot. To achieve this, we require an expression language capable of translating abstract goals into a slot-value style format. We introduce the "Prediction Task Expression Dictionary" (PeTED), consisting of slot/value pairs to define the forecasting task's objectives and constraints.

### 3.1 Prediction Task Expression Dictionary

Assuming a user provides a relational database schema with tables and relations, we can simplify this by treating all tables, whether they define an entity or a relation, as a single joined "giant" table containing all entities and attributes. Figure 1 demonstrates an example schema.

For the purposes of this work, we employ four slots in the PeTED expressions: "Target Attribute", "Aggregation Constraint", "Filter Attribute", and "Filtering Constraint" which can be easily extended to an arbitrary number of slots in the future. The values for "Target Attribute" and "Filter Attribute" can be any attribute from the schema, while "Filtering Constraint" captures constraints such as *equal to* or *greater than*. Meanwhile, "Aggregation Constraint"

| Timestamp | Entities | Variables / Observations |
|-----------|----------|--------------------------|
| $T_1$ | $e_{11}$ , ... , $e_{1k}$ | $o_{11}$ , ... , $o_{1m}$ |
| $T_2$ | $e_{21}$ , ... , $e_{2n}$ | $o_{21}$ , ... , $o_{2p}$ |
| ⋮ | ⋮ | ⋮ |

Figure 1: Sample Database Schema with all entities and attributes joined together.

represents constraints like *count*, *sum*, *average*, etc (see appendix A.2 for details). Consider the following prediction goal and the corresponding *PeTED* expression, for example:

> *For each airline, predict the maximum delay that any of its flights will suffer next week.*

This prediction goal can be expressed by the following *PeTED* expression:

```
Target Attribute: ARRIVAL_DELAY
Filtering Constraint: NONE
Aggregation Constraint: max_agg
```

### 3.2 Filling PeTED Slots

In this section, we discuss the filling process for the *PeTED* slot "Target Attribute". We omit details for other slots due to space constraints, as they follow a similar process. Each PeTED slot is modeled as a random variable, e.g., the *Target Attribute* slot represents a probability distribution over all attributes in a given schema. These probabilities indicate the likelihood of an attribute being the desired *Target Attribute* given a *Forecast Utterance*.

We assume a uniform prior over all attributes, meaning each is initially considered equally likely as the target. Upon receiving a *Forecast Utterance*, the agent extracts information and updates its belief about the *Target Attribute* slot by computing the posterior distribution for the corresponding random variable.

Formally, let the list of attributes in the given schema be $A = \{a_1, a_2, ..., a_n\}$, and $q$ represent the *Forecast Utterance*. Our goal is to rank attributes in $A$ based on $q$. We assume that a user utterance contains clues about the target attribute, so attributes with higher semantic similarity to $q$ are more likely to be the *Target* attribute. Thus, attributes with higher similarity are ranked higher, as they are more likely the desired target attribute.

User utterances are often uncertain and implicit, making it crucial to extract salient phrases for accurate inference. This can be achieved using *Entity Extraction* (EE) techniques (Nasar et al., 2021) or *Question-Answering* (QA) techniques (Soares and Parreiras, 2020), where targeted questions identify slots and answers extract salient phrases. Since a one-to-one mapping from salient phrases to candidate attributes is unlikely, one needs to consider the following two probabilities jointly to make an accurate inference about the *Target Attribute* slot.

1. Given a salient-phrase $x$ extracted from forecast utterance $q$, what is the likelihood that $x$ is indeed relevant to the desired *Target Attribute*? To capture this, we introduce a binary random variable $R_x$, which is defined as the relevance of a salient-phrase with respect to the target attribute.

$$R_x = \begin{cases} 1, & \text{if } x \text{ is a relevant salient-phrase} \\ 0, & \text{otherwise} \end{cases} \tag{1}$$

Here, $x$ can be any salient-phrase extracted from user utterance, such that $x \in Z(q)$, where $Z(q)$ theoretically defines all possible n-grams with $n = \{1, 2, 3, ...\}$. Mathematically, we need to estimate $P(R_x = 1|x)$.

2. Given a relevant salient-phrase $x \in Z(q)$ and an attribute $a_i$ from the data-base schema, what is the probability that $a_i$ is indeed the target attribute? Mathematically, we need estimation of $P(a_i|x, R_x = 1)$.

Finally, all attributes in the data-base schema are ranked according to the following joint probability.

$$P(a_i|q) = \max_{x \epsilon Z(q)} \{P(R_x = 1|x) \times P(a_i|x, R_x = 1)\} \tag{2}$$

### 3.3 The self-supervision with synthetic examples Approach

Calculating probability distributions for *PeTED* representations is challenging due to the non-deterministic nature of user utterances and the varying number of values for each slot in real-time, unseen datasets. Pre-training a model with variable slot options becomes difficult.

To address this, we propose a self-supervised learning approach utilizing synthetic examples for conversational forecasting task formulation. Upon receiving a new dataset, we generate unsupervised, artificial training examples that simulate probable user utterances (explained in Section 4.2). These examples help the model learn the dataset's attributes, granularity, and types. Once trained, the model accurately interprets forecast utterances, as demonstrated by our experiments. With the dataset schema provided, users don't need manual data labeling or annotation, thereby facilitating self-supervised learning with synthetic examples.

## 4 Estimation of PeTED Expressions

In this section, we first outline our assumptions for estimating PeTED expressions. Next, we elaborate on the joint probability estimation process from Equation 2. Finally, we discuss the process of ranking candidate attributes using our probability estimates to fill the *PeTED* slots.

### 4.1 Assumptions

The assumptions for our case studies are as follows:

- Our *PeTED* expression consists of four slots: *Target Attribute*, *Aggregation Constraint*, *Filter Attribute* and *Filtering Constraint*.
- A *Forecast Utterance* may not contain all required information about each slot, i.e., users may provide partial/incomplete information.
- Each slot can have one candidate value.

### 4.2 Estimation of $\mathbf{P(R_x = 1|x)}$

We present the estimation process for the "Target Attribute" slot, while noting that other slots follow a similar approach. The probability $P(R_x = 1|x)$ signifies the likelihood that a salient-phrase $(x)$ extracted from the forecast utterance $(q)$ is relevant to the desired target attribute. Here, $x \in Z(q)$, where $Z(q)$ represents all possible n-grams in $q$, with $n = 1, 2, 3, ....$ Since $Z(q)$ is computationally intractable by definition, we tackle this complexity by utilizing Entity Extraction (EE) and Question Answering (QA) techniques. These methods allow us to extract a limited number of salient-phrases, along with confidence scores, from the forecast utterance and estimate probabilities accordingly.

While EE and QA techniques offer potential in computationally estimating $P(R_x = 1|x)$, directly incorporating a pre-trained EE/QA model is unsuitable for our real-time user-provided database schema scenario. Given the lack of a pre-existing training dataset tailored to each unique schema/domain, fine-tuning pre-trained models is unattainable. With a diverse user base expecting assistance in devising forecasting tasks for their distinct problem domains and database schemas, fine-tuning for every possibility is infeasible.

To address this limitation, we present a robust approach, comprising two methods for synthetic data generation. The first method, a heuristic technique, involves constructing realistic template utterances with empty slots, subsequently populated with relevant attributes and their synonyms derived from the provided schema. This method generates context-specific, custom-crafted examples capturing the core of forecasting tasks. For instance, consider the following template:

<p align="center">Predict the average ___ for each airline tomorrow.</p>

We can fill in the blank of this example template utterance using different attributes and their synonyms from a given user schema.

The second method utilizes a T5 model finetuned on the CommonGen (Lin et al., 2020) task, which generates artificial user utterances via a keyword-to-sequence task, aiming to create more natural-like utterances containing specified slots that partially conform to our templates. We developed three versions of the T5 model: the first remains unaltered, the second is fine-tuned with 1,000 templated samples to integrate template essence subtly, and the third is fine-tuned with 10,000 examples to enforce template structure more emphatically. The resulting T5-based synthetic dataset combines a balanced mixture of utterances from each version, ensuring diversity, various degrees of template conformity, and natural language expression.

We utilize both synthetic datasets along with their corresponding slots to create training samples formatted according to CoNLL-2003 standards (Tjong Kim Sang and De Meulder, 2003) for Entity Extraction (EE) and according to SQuAD standards for Question Answering (QA) (Rajpurkar et al., 2016). This comprehensive foundation allows us to effectively fine-tune pre-trained transformer-based models for EE and QA task. The fine-tuned models are capable of accurately identifying key phrases and generating confidence scores from previously unseen utterances. These confidence scores are instrumental in providing a reliable estimation of the probability $P(R_x = 1|x)$, as demonstrated in our experimental findings.

---

**Algorithm 1:** Forecasting Goal Extraction from User Utterances via Slot-Filling.

| | |
|---|---|
| 1 | **Algorithm** `TaskFormulation()` |
| | **Input:** Attributes $\{a_i\}$, *PeTED*, $utterance(q)$ |
| | **Output:** *PeTED with revised probability distribution* |
| 2 | **Initialization:** pretrained Transformer-based Language Model, *PeTED* with uniform distribution |
| 4 | $D \leftarrow$ TrainingSetGeneration( ) |
| 6 | $model \leftarrow$ Fine-tune Transformer-based LM on EE/QA task using D |
| 8 | $X, X_{conf} \leftarrow$ Salient-Phrases and confidence scores extracted from $q$ by applying $model$ |
| 10 | $P(R_x = 1|x \in X) \leftarrow$ Normalize $X_{conf}$ into a probability distribution |
| 12 | **for** $x \in X$ **do** |
| 14 |     **for** $a_i \in schema$ **do** |
| 16 |         $P(a_i|x, R_x = 1) \leftarrow sem\_sim(a_i, x)$ |
| 17 |     **end** |
| 19 |     Normalize $P(a_i|x, R_x = 1)$ over all $a_i$ |
| 20 | **end** |
| 22 | *PeTED* $\leftarrow$ Re-compute *PeTED* probability distributions using equation 2 |
| 24 | **do** |
| 26 |     Show top item from *PeTED* to user; |
| 27 |     **if** *If user agrees* **then** |
| 28 |         **return** *PeTED*; |
| 29 |     **else** |
| 30 |         Remove top item from *PeTED*; |
| 31 |         Re-compute *PeTED* distributions; |
| 32 |     **end** |
| 33 | **while** *User has not agreed or list is not exhausted* |

---

## 4.3 Estimation of $P(a_i|x, R_x = 1)$

Given a relevant ($R_x = 1$) salient-phrase $x$ and a candidate attribute $a_i$ from the database schema, $P(a_i|x, R_x = 1)$ represents the probability that $a_i$ is the desired target. We assume that attributes with high semantic similarity to relevant salient-phrases are more likely to be the target attribute, as users often mention it directly or through synonyms/salient-phrases. Consequently, we model $P(a_i|x, R_x = 1)$ as proportional to the semantic similarity between $x$ and $a_i$. Mathematically, $P(a_i|x, R_x = 1) \propto Sem\_similarity(a_i, x)$.

## 4.4 Ranking Algorithm for Slot-Filling

Algorithm 1 details the slot-filling process to generate a *PeTED* expression from the user's database schema and forecasting goals. As stated in Section 3.2, the "Target Attribute" values are initialized using a uniform distribution across all schema attributes. Algorithm 1 receives the database schema attributes, initial *PeTED* expression, and

user utterance as input. It then generates an updated *PeTED* expression, incorporating a posterior distribution for the target attribute based on the joint probability calculated from equation 2. The attributes are then ranked based on this probability, with higher values indicating a greater likelihood of being the desired target.

# 5 Case-Studies and Experimental Setup

| Dataset | Model | Entity Extraction | | | | Question Answering | | | |
|---|---|---|---|---|---|---|---|---|---|
| | | Universal | | Custom | | Universal | | Custom | |
| | | Heuristic | T5 | Heuristic | T5 | Heuristic | T5 | Heuristic | T5 |
| **Flight Delay** | Bert | 0.676 | 0.831 | 0.743 | **0.896** | **0.793** | **0.840** | 0.747 | 0.765 |
| | RoBERTa | 0.673 | 0.864 | 0.737 | 0.890 | 0.753 | 0.762 | **0.836** | **0.856** |
| | XLNet | **0.750** | **0.871** | 0.808 | 0.894 | 0.619 | 0.756 | 0.666 | 0.741 |
| | Albert | 0.657 | 0.862 | **0.817** | 0.889 | 0.675 | 0.822 | 0.752 | 0.817 |
| **Online Delivery** | Bert | 0.625 | 0.615 | 0.614 | 0.539 | **0.801** | **0.754** | 0.640 | 0.739 |
| | RoBERTa | 0.637 | 0.609 | 0.619 | **0.582** | 0.729 | 0.735 | **0.760** | **0.741** |
| | XLNet | **0.655** | **0.650** | 0.624 | 0.5709 | 0.633 | 0.606 | 0.536 | 0.595 |
| | Albert | 0.645 | 0.585 | **0.640** | 0.547 | 0.780 | 0.734 | 0.609 | 0.685 |
| **Student Performance** | Bert | 0.569 | 0.539 | **0.553** | 0.525 | **0.610** | 0.56 | **0.639** | **0.570** |
| | RoBERTa | 0.584 | **0.568** | 0.528 | **0.565** | 0.609 | 0.555 | 0.628 | 0.531 |
| | XLNet | **0.595** | 0.521 | 0.515 | 0.469 | 0.582 | 0.516 | 0.472 | 0.529 |
| | Albert | 0.564 | 0.551 | **0.541** | **0.571** | 0.558 | **0.568** | 0.512 | 0.526 |

Table 1: Fine-tuning performance on all four attributes extraction task using EE/QA. *F1* score of models that are trained with **Heuristic**-based and **T5**-based synthetic data are labelled with Heuristic and T5 respectively.

## 5.1 Data-sets and Evaluation Metric

Our methodology stands out by dynamically processing any dataset provided by the user, instantly generating synthetic examples tailored to that dataset. Subsequently, it trains a transformer-based language model (LM), specifically designed to extract information relevant to the dataset directly from user utterances. This streamlined process ensures efficient and accurate retrieval of dataset-specific insights, catering to the immediate needs of users without the prerequisite of pre-established datasets.

In the course of our research, we put our system to the test using three publicly available Kaggle datasets: Flight Delay (FD)[2], Online Food Delivery Preferences (OD)[3] and Student Performance (SP)[4](Details in Appendix A.3). These datasets were chosen to cover a range of topics and data types, showcasing the versatility and robustness of our approach. For evaluating how well our method works, we created test sets for the three datasets using human volunteers with data science expertise, who generated utterances expressing forecasting goals. Each instance consists of a user utterance and associated ground truth slot-value labels. To minimize bias, three volunteers independently created and labeled datasets.

The final test sets contain 344, 170, and 209 sentences for the FD, OD, and SP datasets, respectively. This detailed process ensures that our test sets are both thorough and reliable. To measure how well our model works, we use the F1 score to see how accurately it can identify the correct information in a sentence. We also use the mean reciprocal rank (MRR) to measure the efficacy of our ranking system. These measures help us understand the effectiveness of our model in real-world scenarios, proving the value of our innovative approach in academic research.

## 5.2 Embeddings and Pre-Trained Transformer based Language Models

We employed three word embeddings techniques (Word2Vec Mikolov et al. (2013), GloVe Pennington et al. (2014), FastTextBojanowski et al. (2017) and alongside them we also include Universal Sentence Encoder (USE) Cer et al. (2018)) to capture semantic similarity. The rationale for including a sentence-level approach is that sentence embeddings have demonstrated strong performance in capturing semantic similarity across standard benchmarks Mahajan et al. (2023). We use four pre-trained transformer-based language models (Bert Devlin et al. (2018), RoBERTa Liu et al.

---

[2]https://www.kaggle.com/usdot/flight-delays
[3]https://www.kaggle.com/datasets/benroshan/online-food-delivery-preferencesbangalore-region
[4]https://www.kaggle.com/datasets/larsen0966/student-performance-data-set

(2019), XLNet Yang et al. (2019), and Albert Lan et al. (2020)) to fine-tune for the EE/QA tasks in our case studies. The fine-tuning process for the EE/QA tasks include two variations: 1) **custom** models, fine-tuned exclusively on artificially generated data from a single dataset, and 2) a **universal** model, fine-tuned on artificially generated data for all three datasets. As discussed in Section 4.2, we employed two methods to generate artificial data: heuristic-based and T5-based. In the case study, models fine-tuned using heuristic-generated artificial data are labeled with the keyword **Heuristic**, while those fine-tuned with T5-generated data are labeled with the keyword **T5**.

### 5.3 Hyper-parameter Tuning

Due to the nature of our implementation, our experimental results heavily depend on how well the fine-tuned model (EE/QA) extracts salient-phrases from the user utterances. This, in turn, depends on the effectiveness of the fine-tuning process described in Section 4.2. Therefore, we performed an exhaustive hyperparameter search using a subset of both of our artificially-generated data-set. We then took the highest-scoring set of hyperparameters and fine-tuned the extractive models one more time using the whole training data-set. See Appendix A.6 (Tables 12-15) for details.

## 6 Results

The performance of the fine-tuned extraction models (EE/QA) on the complete training data is displayed in Table 1. By analyzing the results in Table 1, we have identified the preferred model configurations for different settings, as presented in Table 2. We observed that Universal models yield reasonable $F_1$ scores, making them suitable for users with strict memory constraints. Furthermore, T5 models do not outperform Heuristic models across all three test datasets, indicating that if user utterances follow a template like *Forecast Utterance*, training with templated data can be effective.

|           |              | QA      | EE      |
|-----------|--------------|---------|---------|
| Heuristic | **Custom**   | RoBERTa | Albert  |
|           | **Universal**| Bert    | XLNet   |
| T5        | **Custom**   | RoBERTa | RoBERTa |
|           | **Universal**| Bert    | XLNet   |

Table 2: Best extractive models for different settings. Models under the label Heuristic and T5 are the best performing models trained on Heuristic-based and T5-based synthetic data respectively.

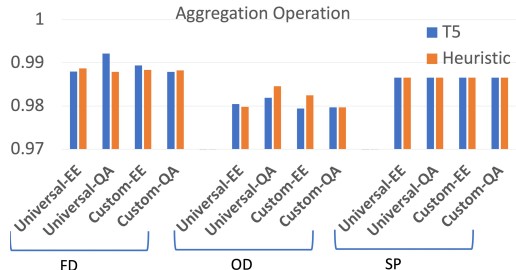
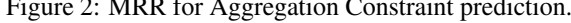

Figure 2: MRR for Aggregation Constraint prediction.

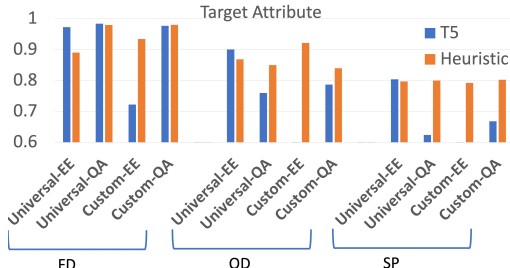

Figure 3: MRR for Target Attribute prediction.

We present the case study results in Figures 2-5 using the recommended configurations from Table 2. The Mean Reciprocal Rank (MRR) for each slot, derived from the four embedding techniques detailed in Section 5.2, is averaged across all models in Table 2. These visualizations demonstrate MRR performance for each slot based on our case study assumptions. Notably, models perform better on datasets with fewer attributes, which is intuitive. Furthermore, the final MRR score is influenced by the model's extraction task performance. In Figure 2, models excel due to the ease of extracting aggregation constraints (count, sum, etc.), while in Figure 5, performance declines as filtering constraints (high, less, etc.) are more abstractive.

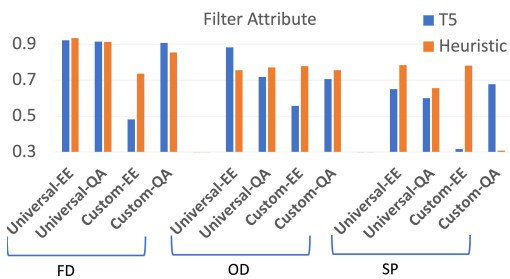
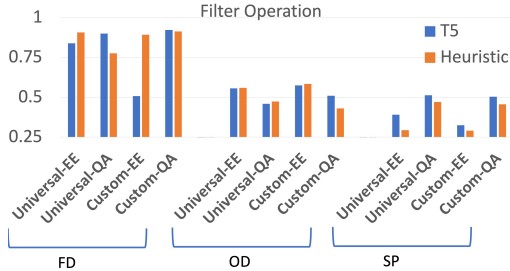

Figure 4: MRR for Filter Attribute prediction.   Figure 5: MRR for Filtering Constraint prediction.

We performed the Wilcoxon Sign-Ranked Test on all model pairs (detailed in Appendix, Table 18) using a flat vector of MRR scores from all datasets and slots. The results revealed statistically significant differences between the MRR scores of the Vanilla method and any other extractive models (EE/QA).

## 6.1 Failure Analysis

In our extractive model, we observe a slower convergence rate when handling utterances with abstract or implicit slot expressions. This is particularly evident in the performance of our proposed T5 and Heuristic models, which struggle to extract filtering constraints (Figure 5) as effectively as they do for other slots (Figures 2-4). Table 3 showcases instances where the Heuristic Universal XLNet model encounters extended convergence times. A closer analysis of these examples reveals that, when users mention slot values implicitly, the model tends to extract incorrect salient phrases, underscoring the difficulties in accurately discerning subtle or abstract expressions within the text.

| User Utterance | Extracted | Count |
|---|---|---|
| predict the total order where the preferred medium of order is online within tomorrow | "within" | 9 |
| Predict the maximum final grade of students where student has extra educational support at school | " " | 7 |

Table 3: This table displays the user utterance (first column), extracted salient phrase (second column), and iterations needed for correct slot-value (third column).

## 6.2 Open-Domain Dialog systems

| User: | Predict the maximum first period grade of students where student has partaken in extra-curricular activities |
|---|---|
| GODEL: | Is this something that can help you plan a schedule for your college students? |
| User: | nope, just formulate a ML problem for the utterance. |
| GODEL: | Do you have any idea on the methods of determining the maximum amount of first period grade of students where student has partaken in extra-curricular activities? |
| User: | nope, that is your job |
| GODEL: | You should get me a job based on this information and that is why I want to get a job. |
| User: | you are not helping with the problem creation. |
| GODEL: | I agree that you are not helping with the problem creation problem. |

Table 4: User Interaction with GODEL. The model is hallucinating and no way of converging.

In this section, we present a qualitative study comparing two recent Open Domain Dialog Systems (ODDS): GODEL and ChatGPT. While a direct comparison between our proposed method and these ODDS is not feasible, we aim to discuss the adaptability of ODDS in the context of this highly ambitious task. Our study encompasses four distinct system setups to evaluate the performance of these models across diverse scenarios. Due to space constraints, we present only one such scenario (see Appendix A.11 for details). For both GODEL and ChatGPT, we provide the following instructions as the system's role: "*You are a Machine Learning expert who will help a small business owner*

| | |
|---|---|
| User: | Predict the maximum first period grade of students where student has partaken in extra-curricular activities |
| ChatGPT: | Can you please tell me the time period for which you want to make this prediction? |
| User: | I do not have time period available with me. Please use the given information to predict. |
| ChatGPT: | To do this, we can consider each school term as a time period and use the following slot values: Aggregation operation: Maximum; Target attribute: First-period grade; Filter attribute: Participation in extracurricular activities; Filter operation: Equals to "Yes" |
| User: | please check the filter operation again. |
| ChatGPT: | Apologies for the mistake. Can you please provide the correct filter operation so that I can update the slot values accordingly? |
| User : | I do not have the filter operation value. Please check the original input to find the value. |
| ChatGPT: | Okay. Just to confirm, did you mean to say that you didn't mention any filter operation value in your original message? |
| User: | No. The filter operation value is mentioned in the original message. Please read the message and find the value. |
| ChatGPT: | The correct slot values are as follows: Aggregation operation: Maximum; Target attribute: First-period grade; Filter attribute: Participation in extracurricular activities; Filter operation: Equals to "Yes" |
| User: | The value of filter operation value is incorrect. |
| ChatGPT: | I apologize for making a mistake earlier. Can you please tell me the correct filter operation value so that I can update the slot values accordingly? |
| User: | I dont know |
| ChatGPT: | Based on your initial request, we can create a time series forecasting problem as follows: Aggregation operation: Maximum; Target attribute: First-period grade; Filter attribute: Participation in extracurricular activities |
| User: | Filter operation value is missing in above response. |

Table 5: User Interaction with ChatGPT. The model struggles with extracting correct filter operation.

*by formulating an interesting problem through conversation. You should perform a slot-filling task based on the conversation and use those slot information to formulate a time series forecasting problem for the user. The slots are aggregation operation, target attribute, filter attribute, and filter operation. You have to infer the slot values from the conversations*". User interactions with GODEL are illustrated in Table 4, while interactions with ChatGPT can be found in Table 5. The study reveals that GODEL fails to converge, while ChatGPT performs relatively well. Both ChatGPT and our proposed model struggle to extract the *Filter operation* constraint. Given the potential of ChatGPT, we plan to explore it as a future direction for improving the extraction of such constraints and enhancing overall model performance. In summary, ChatGPT shows great promise in materializing the vision of "Conversational Data Science".

# 7 Conclusion

In this paper, we have introduced the concept of *Forecast Utterance*, where, a user expresses their forecasting goals through natural language. We have primarily focused on how to automatically understand users' prediction goals from such utterances accurately in an *unsupervised* fashion. As a first step towards the ambitious goal of "Conversational Data Science", we have framed this task as an **unsupervised** slot-filling problem and, accordingly, proposed **self-supervised solutions with synthetic examples** based on Entity Extraction(EE)/Question Answering(QA) techniques. Due to the task's user-centric and real-time nature, the lack of custom training data for fine-tuning EE/QA models in the ranking algorithm was a significant challenge, which we addressed by our proposed heuristic and T5 based synthetic data generation technique. Experimental results show that the proposed synthetic data generation technique significantly improved the accuracy of the overall ranker, and, therefore, demonstrates the feasibility of a general "Conversational

Data Science" paradigm. This task is ambitious as well as multidisciplinary in nature and requires attention from multiple research communities, including NLP, AutoML and HCI.

## 8 Limitations

Our work is limited by the following assumptions made in the paper:

- We assumed that our *PeTED* expression consists of four slots: *Target Attribute*, *Aggregation Constraint*, *Filter Attribute* and *Filtering Constraint*, which may not hold in some real word scenarios.
- In this work, we assumed that a *Forecast Utterance* may not contain all required information about each slot, i.e., users may provide partial/incomplete information. But, we have not studied how to prompt users to gather further information or ask clarification questions, which is an interesting future work.
- We have assumed that each slot can have one value at maximum, which may not hold in real word. We plan to extend our approach to multiple values scenario in the future.

Although our case study is limited in many ways, this limited scope should not undermine the potential impact of our proposed idea, i.e., *Forecasting Task Formulation Via Natural Conversation*. Imagine Alexa, Siri, Watson and other home assistants serving as your personal data scientists. As the first work of its kind, we made a choice regarding the inevitable trade-off between taking a small step towards an ambitious goal vs taking a significant step towards solving a well-defined problem. Thus, we had to leave more complex scenarios as future work. As a novel perspective, we believe our work will inspire the researchers to pursue this ambitious problem and make substantial impact through wider adoption of such "end-user-in-the-loop" conversational AutoML solutions.

## 9 Acknowledgements

This work has been partially supported by the National Science Foundation (NSF) Standard Grant Award #2302974 and Air Force Office of Scientific Research Grant/Cooperative Agreement Award #FA9550-23-1-0426.

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

# A  Appendix

---

**Algorithm 2:** Artificial Training Data Generation through Heuristic for Fine-tuning EE/QA Models for "Target Attribute" slot.

1 **Algorithm** `TrainingDataWithHeuristic()`
   **Input:** Schema $S$, Template utterance set $T$, Attribute-phrases $A$
   **Output:** Artificial Training Dataset
2    $U \leftarrow$ Generate artificial utterances by filling every $t \in T$ with every $a \in A$.
3    $U_{syn} \leftarrow$ Create additional artificial utterances from set $U$ by replacing every $a \in A$ with their synonyms.
4    $U \leftarrow U \cup U_{syn}$.
5    $U_L \leftarrow$ annotate each $u \in U$ with salient-phrases labeled as the target attribute.
6    return $U_L$

---

**Algorithm 3:** Generating K synthetic utterances using T5 model variants based on attributes: The algorithm takes attributes, a T5 model, templates, and K as input, and generates a synthetic dataset containing K balanced utterances from three T5 model variants, each with different levels of template conformity.

**Input:** Attributes, T5 model, templates, K
**Output:** Synthetic dataset
1 Load T5 model finetuned on CommonGen task
2 Initialize synthetic dataset $D \leftarrow \{\}$
3 **for** $i \in \{1, 2, 3\}$ **do**
4    **if** $i == 1$ **then**
5       $M \leftarrow$ T5 model (unaltered)
6    **else if** $i == 2$ **then**
7       $M \leftarrow$ T5 model fine-tuned with 1,000 templated samples
8    **else**
9       $M \leftarrow$ T5 model fine-tuned with 10,000 examples
10   **end**
11   Generate $\frac{K}{3}$ artificial user utterances $U_i$ using $M$ and attributes for keyword-to-sequence task
12   $D \leftarrow D \cup U_i$
13 **end**
14 Mix utterances in $D$ to create a balanced blend of utterances from each model variant
15 **return** $D$

## A.1 Artificial Data Generation

We present a robust approach, comprising two methods for artificial data generation. As described in Algorithm 2 the first method, a heuristic technique, involves constructing realistic template utterances with empty slots, subsequently populated with relevant attributes and their synonyms derived from the provided schema.

The second method as described in Algorithm 3, utilizes a T5 model fine-tuned on CommonGen (Lin et al., 2020) task. The CommonGen task, as detailed in Lin et al. (2020), is designed to enhance text generation models like T5 by requiring them to create coherent sentences from a set of given words. This task evaluates the model's ability to weave these words into contextually meaningful sentences, testing its comprehension and creative synthesis skills. Fine-tuning T5 on CommonGen specifically aims to improve its generative commonsense reasoning, making it adept at producing human-like, contextually rich text. In our study, we used a T5 model that had been specially trained with the CommonGen task to generate synthetic user sentences. This approach involves transforming a list of keywords into full sentences that sound natural and follow certain patterns we have set. This is aimed at creating sentences that include specific information (or 'slots') in a way that makes sense and is useful for our research. For example, when we input the keywords "MAX", "DURATION_OF_SCHEDULED_TRIP", "DESTINATION_AIRPORT", "CANCELLATION_REASON", "IS_MORE_THAN" into our trained T5 model, it generated the following sentence:

"Find the longest scheduled trip to each destination airport where the reason for cancellation was given more than one day earlier."

This example shows how the model can take a list of keywords and turn them into a clear, useful sentence. It highlights the model's ability to understand and use language in a way that matches how people might ask for information, demonstrating its value for creating better automated conversation systems.

### A.1.1 Artificial Data Formats

After We generate the artificial dataset, we convert them into two format for training the Transformer Based Models. Each Format design is described below:

**CoNLL-2003 standard**: In this format we annotate each token of the generated data with appropriate labels. Our customized labels are: I-ATR (Target Attribute), I-AGG (Aggregation operator), I-FLT (Filtering Attribute), I-FLO (Filtering Operator). For example, If the generated sentence is "Predict the average departure delay for each airline wherre elapsed time is more than five hours for tomorrow." Then the annotated data will be as following:

```
Predict NN I-NP O
the NN I-NP O
average NN I-NP I-AGG
departure NN I-NP I-ATR
delay NN I-NP I-ATR
for NN I-NP O
each NN I-NP O
airline NN I-NP O
where NN I-NP O
elapsed NN I-NP I-FLT
time NN I-NP I-FLT
is NN I-NP O
more NN I-NP I-FLO
than NN I-NP I-FLO
five NN I-NP O
hours NN I-NP O
for NN I-NP O
tomorrow NN I-NP O
```

Again if the artificial data is "Predict the average first period grade of students where school is Mousinho da Silveira" then the annotated data will look like the following:

```
Predict NN I-NP O
the NN I-NP O
average NN I-NP I-AGG
```

```
first NN I-NP I-ATR
period NN I-NP I-ATR
grade NN I-NP I-ATR
of NN I-NP O
students NN I-NP I-ENT
where NN I-NP O
school NN I-NP I-FLT
is NN I-NP I-FLO
Mousinho NN I-NP O
da NN I-NP O
Silveira NN I-NP O
```

**SQuAD standards for Question Answering**: For the SQuAD format, we consider the artificial data as the context for this task and we try to answer slot specific question from it. We use the following questions:

**Aggregator**: what is the target aggregator?

**Target Attribute**: what is the target attribute?

**Filter Attribute**: what is the target filtering attribute?

**Filter Operator**: what is the target filtering operator?

Examples are as follows:

```
1: {
  "context": "predict the average departure delay for each airline tomorrow",
  "qas": [
    {
      "question": "what is the target aggregator?",
      "answers": [
        { "text": "average", "answer_start": 12, "answer_end": 19 }
      ],
      "id": "..."
    },
    {
      "question": "what is the target attribute?",
      "answers": [
        {
          "text": "departure delay",
          "answer_start": 20,
          "answer_end": 35
        }
      ],
      "id": "..."
    }
  ]
}

2: {
    "context": "predict the average first period grade in the next fall semester among the students
                where school is mousinho da silveira",
    "qas": [
        {
          "question": "what is the target aggregator?",
          "answers": [
            { "text": "average", "answer_start": 12, "answer_end": 19 }
          ],
          "id": "..."
        },
        {
```

```
        "question": "what is the target attribute?",
        "answers": [
          {
            "text": "first period grade",
            "answer_start": 20,
            "answer_end": 38
          }
        ],
        "id": "..."
      },
      {
        "question": "what is the target filtering attribute?",
        "answers": [
          { "text": "school", "answer_start": 90, "answer_end": 96 }
        ],
        "id": "..."
      },
      {
        "question": "what is the target filtering operator?",
        "answers": [
          { "text": "is", "answer_start": 97, "answer_end": 99 }
        ],
        "id": "..."
      }
    ]
}
```

## A.2   Filtering and Aggregation Operations

We have designed *PeTED* (Section 3.1) in such a way that it will provide constructs for variables, operators, and functions. Note that the type of variables is dependent on implementation, and the operators will depend on specific variable types. We have included some basic operators, which include filtering operators (equal to, greater than, etc.) and aggregation operators (count, sum, average, etc.) as part of this case study. Table 6 shows a list of sample operators for our ***Forecast Utterance Understanding*** agent, along with their supported types.

## A.3   Datasets

We experimented with three datasets: Flight Delay (FD), Online Food Delivery Preferences (OD), and Student Performance (SP) Dataset. Tables 7-9 shows the attributes and entities of those three dataset schemas. Using these datasets, we conducted an exploratory study to understand how our proposed slot-filling method performs in a practical scenario. We use Algorithm 2 to generate artificial data for each attribute of the used datasets. We format the generated artificial training examples into the CoNLL-2003 format for the EE task and the SQuAD format for the QA task. We split those data into an 8:2 ratio for training and testing split. We have created handcrafted validation sets for each of the three datasets by actively engaging human volunteers with decent data science expertise and asking them to create utterances expressing forecasting goals. Along with user utterances, each instance in the validation set contains ground truth labels correlated to it. We engaged three volunteers to create and label the datasets independently to avoid bias. Additionally, we cross-validated the dataset created by one volunteer by scrutinizing it with another volunteer, not involved in generating the particular dataset in the first phase. The handcrafted validation data-sets contain 344, 170 and, 209 utterances for **FD**, **OD** and **SP** data-set, respectively.

### A.3.1   Qualitative Examples

In this section, we showcase brief and precise explanations several utterances selected from our handcrafted validation dataset and describe the challenges to extract time series formulation related slot value from them.

- **Utterance**: "I want to know the total departure delay for each airline that will be delayed more than five minutes by air system next week."

  **Domain**: Flight Delay

  **Description**: This example represents a time series forecasting problem in airline operations. The goal is to predict the total departure delay for each airline with air system delays over five minutes next week. A model can be trained on historical data, capturing time-dependent patterns to optimize resource planning and minimize passenger impact.

  **Challenges**:

  - **Implicit information**: Some critical information for problem formulation is not explicitly mentioned, such as the desired granularity of the forecast (e.g., daily, hourly) or the specific features to be used in the model.
  - **Complex relationships**: The utterance refers to multiple interrelated variables ("total departure delay" and "airlines with delays more than five minutes") that need to be understood in the context of the problem. This adds complexity to the task of extracting relevant slot-values.
  - **Multiple slot-values**: The utterance combines several aspects of the problem, which can make it challenging to extract and separate the relevant slot-values accurately.
  - **Domain-specific knowledge**: Understanding the context of the utterance requires domain-specific knowledge of the airline industry and the factors that contribute to flight delays. This knowledge is necessary to identify and extract the correct slot-values.

- **Utterance**: "Predict the average change in duration of flights for Qatar Airways which will have an elapsed time more than four hours and will start within the next week".

  **Domain**: Flight Delay

  **Description**: This example represents a time series forecasting problem in airline operations. The goal is to forecast the average change in flight duration for Qatar Airways for flights with over four hours of elapsed time starting in the next week. A model can be trained on historical data to capture time-dependent trends, helping airlines optimize scheduling and enhance customer experience.

  **Challenges**:

  - **Ambiguity**: The utterance contains terms like "average change in duration" that might be difficult to map directly to specific slot-values without clarification on whether it refers to an increase or decrease in duration, or both.
  - **Complex relationships**: The utterance refers to multiple interrelated variables ("elapsed time more than four hours" and "flights starting within the next week") that need to be understood in the context of the problem. This adds complexity to the task of extracting relevant slot-values.
  - **Multiple slot-values**: The utterance combines several aspects of the problem, making it challenging to extract and separate the relevant slot-values accurately.
  - **Domain-specific knowledge**: Understanding the context of the utterance requires domain-specific knowledge of the airline industry and the factors that contribute to changes in flight duration. This knowledge is necessary to identify and extract the correct slot-values

- **Utterance**: "Predict the maximum order placed by mistake where the age of a customer is less than 21 for the next month".

  **Domain**: Online Delivery

  **Description**: This example represents a time series forecasting problem in the online delivery domain. The objective is to forecast the maximum number of mistaken orders placed by customers under 21 years old for the upcoming month. By training a model on historical data and capturing time-dependent trends, businesses can proactively identify and address potential issues, improving customer service and satisfaction.

  **Challenges**:

  - **Complex relationships**: The utterance refers to multiple interrelated variables ("maximum order placed by mistake" and "age of a customer less than 21") that need to be understood in the context of the problem. This adds complexity to the task of extracting relevant slot-values.

- **Multiple slot-values**: The utterance combines several aspects of the problem, making it challenging to extract and separate the relevant slot-values accurately.
- **Domain-specific knowledge**: Understanding the context of the utterance requires domain-specific knowledge of the order placement process and factors that contribute to mistaken orders. This knowledge is necessary to identify and extract the correct slot-values.

- **Utterance**: "Predict the total order where the preferred medium of order is online within tomorrow".

  **Domain**: Online Delivery

  **Description**: This example represents a time series forecasting problem in the online delivery domain. The goal is to predict the total number of orders placed online for tomorrow. A model trained on historical data can capture time-dependent trends, helping businesses prepare for demand and optimize their operations.

  **Challenges**:

  - **Complex relationships**: The utterance refers to a single variable ("total order where the preferred medium of order is online"), which needs to be understood in the context of the problem. This adds complexity to the task of extracting relevant slot-values.
  - **Multiple slot-values**: The utterance combines several aspects of the problem, making it challenging to extract and separate the relevant slot-values accurately.
  - **Domain-specific knowledge**: Understanding the context of the utterance requires domain-specific knowledge of the order placement process and factors that contribute to online orders. This knowledge is necessary to identify and extract the correct slot-values.

- **Utterance**: "Predict the maximum final grade of students where the student has extra educational support at school".

  **Domain**: Student Performance

  **Description**: This example represents a time series forecasting problem in the student performance domain. The objective is to predict the maximum final grade for students who receive extra educational support at school. By training a model on historical data and capturing time-dependent trends, schools can better understand the impact of additional support and adapt their educational strategies accordingly.

  **Challenges**:

  - **Target variable**: The utterance mentions "maximum final grade", which is a single variable. However, it is not clear whether the goal is to predict the maximum final grade for an individual student or for a group of students with extra educational support. This ambiguity adds complexity to the task of extracting relevant slot-values.
  - **Data filters**: The utterance introduces a filtering condition based on extra educational support. Extracting this filter accurately is crucial to ensure that the problem is well-formulated and the predictions are relevant.
  - **Domain-specific knowledge**: Understanding the context of the utterance requires domain-specific knowledge of the educational system and the factors that contribute to student performance. This knowledge is necessary to identify and extract the correct slot-values, such as the features to be used in the model.

The qualitative study examines various time series forecasting problem formulations across different domains, emphasizing the unique challenges in extracting slot-values from user utterances.

**Flight Delay Domain**: In this domain, challenges include handling implicit information not explicitly mentioned in the utterance, addressing ambiguity in terminology, understanding complex relationships between variables, extracting multiple slot-values, and utilizing domain-specific knowledge about the airline industry and factors that contribute to flight delays.

**Online Delivery Domain**: For this domain, challenges comprise understanding complex relationships between variables, such as order placement processes and factors contributing to mistaken orders, extracting multiple slot-values, and incorporating domain-specific knowledge about online delivery and e-commerce.

**Student Performance Domain**: In the context of student performance, challenges involve dealing with ambiguity in the target variable (e.g., individual or group performance), extracting data filters accurately, such as identifying specific conditions like extra educational support, and applying domain-specific knowledge about the educational system and factors contributing to student performance.

Addressing these challenges requires the development of advanced natural language processing techniques, context-aware algorithms, and incorporating domain knowledge to accurately extract slot-values and formulate time series forecasting problems. Each domain presents unique difficulties, making it essential to adapt models and techniques to handle specific domain-related complexities effectively.

Table 6: Operators

| Operation Set | Supported Ops | Supported Types |
|---|---|---|
| Filter $\mathbf{O}_f$ | `all_fil` | None |
| | `greater_fil`, `less_fil` | Numerical |
| | `eq_fil`, `neq_fil` | Categorical/Entity |
| Aggregation $\mathbf{O}_g$ | `count_agg` | None |
| | `sum_agg`, `avg_agg`, | Numerical |
| | `min_agg`, `max_agg` | Numerical |
| | `majority_agg` | Categorical/Entity |

Table 7: Flight Delay Data-set details

| Attribute Type | Attribute Name and Meanings |
|---|---|
| Timestamp | Date, Day_of_week scheduled_departure_hour scheduled_time, elapsed_time |
| Entities | Airline, Flight_number, Tail_number Origin_airport, Destination_airport |
| Categorical | Cancelled_status, Cancellation_reason |
| Numerical | Departure_delay, Arrival_delay, Airline_delay, System_delay, Security_delay, Late_aircraft_delay, Weather_delay |

Table 8: Online Food Delivery Data-set details

| Attribute Type | Attribute Name and Meanings |
|---|---|
| Timestamp | Order Time |
| Entities | Food Order |
| Categorical | Gender, Marital Status, Occupation, Educational Qualifications, Medium, Meal, Preference, Time saving, Easy Payment option, More Offers and Discount, Good Tracking system, Self Cooking, Health Concern, Late Delivery, Poor Hygiene, Unavailability, Unaffordable, Influence of time, Residence in busy location, Google Maps Accuracy, Good Road Condition, Low quantity low time, Delivery person ability, High Quality of package, Politeness, Freshness, Good Taste, Good Quantity, |
| Numerical | Monthly Income, Family size, latitude, longitude, Pin code, Ease and convenient, More restaurant choices, Good Food quality, Bad past experience, Long delivery time, Delay of delivery person getting assigned, Delay of delivery person picking up food, Wrong order delivered, Missing item, Order placed by mistake, Maximum wait time, Influence of rating, Less Delivery time, Number of calls, Temperature, Reviews |

## A.4 Transformer-based Language Models

We used existing Transformer-based Language Models and fine-tuned them on Entity Extraction and Question Answering tasks to extract salient phrases related to the target slots from user utterances to formulate the goal prediction task.

Table 9: Student Performance Data-set details

| Attribute Type | Attribute Name and Meanings |
| --- | --- |
| Binary | Student's school, Student's sex, Student's home address type, Family size, Parent's cohabitation status, Extra educational support, Wants to take higher education, Internet access at home, With a romantic relationship, Family educational support, Extra paid classes within the course subject, Extra-curricular activities, Attended nursery school, |
| Entity | Students |
| Nominal | Mother's job, Father's job, Student's guardian |
| Numerical | First period grade, Second period grade, Final grade, Free time after school, Going out with friends, Workday alcohol consumption, Weekend alcohol consumption, current health status, Number of school absences, Quality of family relationships, Number of past class failures, Weekly study time, Home to school travel time, Father's education, Mother's education, Student's age |

We have used four popular Transformer-based Language Models for our case study in addition to a baseline method without using any Language Models.

**Bert:** Bert (Bidirectional Encoder Representations from Transformers) Devlin et al. (2018) has delivered a phenomenal impact in the Machine Learning community, achieving state-of-the-art results in a wide variety of NLP tasks, such as Question Answering (SQuAD v1.1), Natural Language Inference (MNLI), Named Entity Recognition, and so on. We use the 'bert-base-cased' variation of the model which contains 110 million parameters.

**RoBERTa**: RoBERTa Liu et al. (2019) is an extension of BERT with several changes to the pre-training procedure, including training for longer time periods, with larger batches, over more data, and without the next sentence prediction objective. We use 'roberta-base' variation of the model which contains 123 million parameters.

**XLNet**: XLNet Yang et al. (2019) is an auto-regressive language model which outputs the joint probability of a sequence of tokens based on the transformer architecture with recurrence. XLNet is an extension of the Transformer-XL model and has also performed well in EE and QA tasks. We use 'xlnet-base-cased' variation of the model which contains 110M parameters.

**Albert**: Albert Lan et al. (2020) introduce A Lite Bert architecture that incorporates factorized embedding parameterization and cross-layer parameter sharing parameter-reduction techniques. We use the base version of Albert ('albert-base-v2') which contains 31 million parameters.

### A.5 Word-Embedding Techniques

We use phrase embeddings as part of our process to compute the semantic similarity (cosine and euclidean distance) between the user utterances and candidate target attributes. The list of attributes in the dataset are ranked according to these semantic similarity scores. In this case study, we used three state-of-the-art word embeddings and one sentence enbedding.

**Word2Vec:** Word2Vec Mikolov et al. (2013) is a way to vectorize text. It's a two-layer neural net which vectorizes words. For input, it takes a text corpus and it outputs a set of vectors. These vectors are called feature vectors, which represent words in the corpus used. We used a pre-trained version of Word2Vec provided by Google. This embedding is trained on the Google News dataset, and the model contains 300-dimensional vectors for 3 million words and phrases.

**FastText:** FastTextBojanowski et al. (2017) is a word embedding method where each word is represented as an n-gram of characters. our used version of FastText that contains 1 million word vectors trained on the Wikipedia 2017, UMBC webbase corpus, and statmt.org news dataset (16B tokens) .

**GloVe:** GloVe Pennington et al. (2014) is another popular embedding technique which is trained on aggregated global word-word co-occurrence statistics from a corpus. We used a version of GloVe embedding that contains 6B tokens, has a vocab size of 400K, is uncased, and is 300-dimensional. it is trained on the Wikipedia 2014 dump alongside the English Gigaword 5th edition dataset.

**Universal Sentence Encoder:** The Universal Sentence Encoder encodes text into high dimensional vectors that can be used for text classification, semantic similarity, clustering, and other natural language tasks.

## A.6 Hyper-Parameter Tuning

We performed an exhaustive hyperparameter search using a subset of the artificially generated dataset. We present the search space for out hyperparameter tuning in Table 10 and Table 11, where we varied the learning rate and weight decay. We choose the values of learning rate and weight decay by manual tuning. We perform the trial of hyperparameter tuning once, and $F_1$ scores of extractive models using the handcrafted validation dataset are monitored during the trial. We then took the highest-scoring set of hyperparameters and fine-tuned the extractive models one more time using the whole training dataset. In Table 12 and Table 13 we present the final set of hyperparameters selected for Entity extraction and Question Answering tasks respectively using the Heuristic based dataset. In Table 14 and Table 15 we present the final set of hyperparameters selected for Entity extraction and Question Answering tasks respectively using the T5 based dataset.

| Hyperparameter | Search space |
|---|---|
| number of epochs | 1 |
| batch size | 64 |
| learning rate | $[1e-6, 1e-4]$ |
| Weight decay | $[0.01, 0.3]$ |
| learning rate optimizer | Adam |
| adam epsilon | 1e-8 |
| max sequence length | 64 |

Table 10: Hyperparameter search space for Entity Extraction models

| Hyperparameter | Search space |
|---|---|
| number of epochs | 1 |
| batch size | 32 |
| learning rate | $[1e-6, 1e-4]$ |
| Weight decay | $[0.01, 0.3]$ |
| learning rate optimizer | Adam |
| adam epsilon | 1e-8 |
| max sequence length | 128 |
| max answer length | 30 |

Table 11: Hyperparameter search space for Question Answering models

| Methods | Flight Delay | | Online Delivery | | Student Performance | | Universal | |
|---|---|---|---|---|---|---|---|---|
| | Weight Decay | Learning Rate | Weight Decay | Learning Rate | Weight Decay | Learning Rate | Weight Decay | Learning Rate |
| BERT | 0.25 | 7.5e-5 | 0.01 | 7.5e-5 | 0.1 | 1e-5 | 0.15 | 5e-5 |
| XLNet | 0.01 | 7.5e-5 | 0.15 | 1e-4 | 0.3 | 1e-4 | 0.05 | 7.5e-5 |
| RoBERTa | 0.25 | 1e-5 | 0.25 | 1e-5 | 0.2 | 1e-5 | 0.3 | 5e-5 |
| Albert | 0.01 | 5e-5 | 0.3 | 1e-5 | 0.05 | 1e-4 | 0.15 | 1e-5 |

Table 12: Final Model Hyper-parameters for Question Answering task using Heuristic based artificial data

## A.7 Experimental Setups

In Table 16, we present the estimated time to fine-tune all combination of models we present in the case study. We have used **one** *Nvidia Quadro RTX 5000* GPU and reported the time needed to fine-tune the models in our case study. As universal models use artificial data for all three dataset, we present them in first four rows in Table 16. As the ensemble models uses fine-tuned EE and QA models' output and perform ensemble operation on them, we do not specifically fine-tune those models.

| Methods | Flight Delay Weight Decay | Flight Delay Learning Rate | Online Delivery Weight Decay | Online Delivery Learning Rate | Student Performance Weight Decay | Student Performance Learning Rate | Universal Weight Decay | Universal Learning Rate |
|---|---|---|---|---|---|---|---|---|
| BERT | 0.25 | 7.5e-5 | 0.2 | 5e-4 | 0.3 | 5e-5 | 0.1 | 7.5e-5 |
| XLNet | 0.1 | 5e-5 | 0.01 | 7.5e-5 | 0.01 | 7.5e-5 | 0.3 | 5e-5 |
| RoBERTa | 0.25 | 5e-5 | 0.15 | 1e-4 | 0.15 | 1e-4 | 0.25 | 7.5e-5 |
| Albert | 0.1 | 5e-5 | 0.3 | 1e-5 | 0.3 | 5e-5 | 0.3 | 1e-4 |

Table 13: Final Model Hyper-parameters for Entity Extraction task using Heuristic based artificial data

| Methods | Flight Delay Weight Decay | Flight Delay Learning Rate | Online Delivery Weight Decay | Online Delivery Learning Rate | Student Performance Weight Decay | Student Performance Learning Rate | Universal Weight Decay | Universal Learning Rate |
|---|---|---|---|---|---|---|---|---|
| BERT | 0.3 | 7.5e-5 | 0.05 | 5e-5 | 0.2 | 0.0005 | 0.3 | 0.0005 |
| XLNet | 0.01 | 0.0001 | 0.25 | 5e-5 | 0.15 | 5e-5 | 0.3 | 7.5e-5 |
| RoBERTa | 0.25 | 5e-5 | 0.15 | 1e-5 | 0.3 | 5e-5 | 0.01 | 0.0001 |
| Albert | 0.25 | 0.0001 | 0.3 | 1e-5 | 0.3 | 7.5e-5 | 0.01 | 7.5e-5 |

Table 14: Final Model Hyper-parameters for Question Answering task using T5 generated artificial data

## A.8 Reproducibility

**For all reported experimental results:**

| Methods | Flight Delay Weight Decay | Flight Delay Learning Rate | Online Delivery Weight Decay | Online Delivery Learning Rate | Student Performance Weight Decay | Student Performance Learning Rate | Universal Weight Decay | Universal Learning Rate |
|---|---|---|---|---|---|---|---|---|
| BERT | 0.05 | 7.5e-5 | 0.2 | 7.5e-5 | 0.01 | 0.0001 | 0.05 | 7.5e-5 |
| XLNet | 0.3 | 7.5e-5 | 0.25 | 7.5e-5 | 0.05 | 0.0001 | 0.1 | 7.5e-5 |
| RoBERTa | 0.15 | 0.0001 | 0.05 | 0.0001 | 0.1 | 7.5e-5 | 0.25 | 0.0001 |
| Albert | 0.05 | 7.5e-5 | 0.01 | 7.5e-5 | 0.15 | 0.0001 | 0.25 | 5e-5 |

Table 15: Final Model Hyper-parameters for Entity Extraction task using T5 generated artificial data

| Dataset | Model | Entity Extraction Universal Heuristic | Entity Extraction Universal T5 | Entity Extraction Custom Heuristic | Entity Extraction Custom T5 | Question Answering Universal Heuristic | Question Answering Universal T5 | Question Answering Custom Heuristic | Question Answering Custom T5 |
|---|---|---|---|---|---|---|---|---|---|
| **FD** | Bert | 105 | 120 | 100 | 122 | 605 | 780 | 570 | 792 |
| | RoBERTa | 90 | 123 | 105 | 124 | 600 | 820 | 540 | 804 |
| | XLNet | 160 | 150 | 150 | 155 | 1860 | 2797 | 1800 | 2889 |
| | Albert | 120 | 140 | 120 | 137 | 610 | 801 | 590 | 817 |
| **OD** | Bert | - | - | 100 | 121 | - | - | 675 | 1248 |
| | RoBERTa | - | - | 104 | 123 | - | - | 700 | 1996 |
| | XLNet | - | - | 157 | 154 | - | - | 2160 | 4082 |
| | Albert | - | - | 121 | 135 | - | - | 690 | 1251 |
| **SP** | Bert | - | - | 103 | 121 | - | - | 555 | 859 |
| | RoBERTa | - | - | 100 | 123 | - | - | 560 | 856 |
| | XLNet | - | - | 160 | 159 | - | - | 1800 | 2781 |
| | Albert | - | - | 125 | 136 | - | - | 585 | 837 |

Table 16: Time required (in seconds) for fine-tuning on all four attributes extraction task using EE/QA and two variations of Ensemble methods.

- A clear description of the mathematical setting, algorithm, and/or model: Mathematical design and problem formulation is discussed in Section 3.2. We have discussed how we do the artificial dataset generation in Algorithm 2, and the process of slot filling with user in the end loop in Algorithm 1.

- Submission of a zip file containing source code, with specification of all dependencies, including external libraries, or a link to such resources (while still anonymized) Description of computing infrastructure used: A zip file will be submitted in the submission panel along with the manual.

- The average runtime for each model or algorithm (e.g., training, inference, etc.), or estimated energy cost: Average runtime is discussed in Appendix A.7.

- Number of parameters in each model: parameter count of each model we used is discussed in Appendix A.4

- Corresponding validation performance for each reported test result: We artificially generate data to train out extractive model and test those model on handcrafted dataset described in Section 5.1. As creation of handcrafted dataset is very costly so, we have used all of human generated dataset as out test set hence, we do not have a validation dataset for our case study.

- Explanation of evaluation metrics used, with links to code: Explanation of evaluation metric used is given in Section 5.1. Link to code is not applicable as we implement it ourself. This code will be included inside the zip file that we will submit in the paper submission panel.

**For all experiments with hyperparameter search:**

- The exact number of training and evaluation runs: details in Appendix A.6.

- Bounds for each hyperparameter: details in Appendix A.6

- Hyperparameter configurations for best-performing models: details in Appendix A.6

- Number of hyperparameter search trials: details in Appendix A.6

- The method of choosing hyperparameter values (e.g., uniform sampling, manual tuning, etc.) and the criterion used to select among them (e.g., accuracy): details in Appendix A.6

- Summary statistics of the results (e.g., mean, variance, error bars, etc.): details in Appendix A.6

**For all datasets used:**

- Relevant details such as languages, and number of examples and label distributions: details in Appendix A.3

- Details of train/validation/test splits: details in Appendix A.3

- Explanation of any data that were excluded, and all pre-processing steps: details in Appendix A.3

- A zip file containing data or link to a downloadable version of the data: Not applicable, as we generate data artificially.

- For new data collected, a complete description of the data collection process, such as instructions to annotators and methods for quality control: details in Appendix A.3

### A.9 Statistical Significance Tests

We also report the average of MRR numbers across all embeddings and present them in Table 17, where we can clearly see that extractive models works better than **vanilla** model (without using any extraction) in almost all the cases. We further conducted the Wilcoxon Sign-Ranked Test for all pairs of models (presented in appendix, Table 18). We took the MRR score for all the data-sets and slots and concatenated them into a flat vector for each model to conduct the statistical analysis. We found that the observed difference between the MRR scores of Vanilla method and any other extractive model (EE/QA) are statistically significant.

| Models | | Aggregator | | Target Attribute | | | Filter Operation | | | Filter Attribute | | | |
|--------|--------|-----|-----|-----|-----|-----|-----|-----|-----|-----|-----|-----|-----|
| | | FD | OD | SP | FD | OD | SP | FD | OD | SP | FD | OD | SP |
| | Vanilla | 0.541 | 0.667 | 0.845 | 0.37 | 0.342 | 0.416 | 0.480 | 0.456 | 0.387 | 0.437 | 0.246 | 0.483 |
| Heuristic | XLNet | **0.988** | 0.979 | **0.986** | 0.890 | 0.86 | 0.797 | 0.906 | 0.559 | 0.293 | **0.933** | 0.754 | **0.783** |
| | Bert | 0.987 | **0.984** | **0.986** | 0.979 | 0.849 | 0.799 | 0.775 | 0.473 | **0.470** | 0.912 | 0.770 | 0.65 |
| | RoBERTa | **0.988** | 0.979 | **0.986** | **0.98** | 0.839 | **0.802** | **0.913** | 0.430 | 0.455 | 0.854 | 0.755 | 0.307 |
| | Albert | **0.988** | 0.982 | **0.986** | 0.934 | **0.921** | 0.792 | 0.892 | **0.584** | 0.290 | 0.73 | **0.777** | 0.780 |
| T5 | XLNet | 0.987 | 0.980 | 0.987 | 0.973 | 0.901 | 0.804 | 0.839 | 0.556 | 0.391 | 0.921 | 0.882 | 0.649 |
| | Bert | 0.992 | 0.982 | 0.987 | 0.984 | 0.759 | 0.624 | 0.900 | 0.459 | 0.512 | 0.914 | 0.718 | 0.599 |
| | RoBERTa-EE | 0.989 | 0.979 | 0.987 | 0.722 | 0.313 | 0.433 | 0.508 | 0.574 | 0.325 | 0.481 | 0.556 | 0.316 |
| | RoBERTa-QA | 0.988 | 0.980 | 0.987 | 0.977 | 0.787 | 0.669 | 0.922 | 0.509 | 0.504 | 0.907 | 0.705 | 0.677 |

Table 17: Average MRR of all model/embeddings.

| | | | Heuristic | | | | T5 | | | |
|--------|--------|---------|--------|--------|---------|--------|------------|------------|--------|--------|
| | | vanilla | XLNet | Bert | RoBERTa | Albert | RoBERTa-EE | RoBERTa-QA | XLNet | Bert |
| | vanilla | 1.0 | - | - | - | - | - | - | - | - |
| Heuristic | XLNet | 9.84e-9 | 1.0 | - | - | - | - | - | - | - |
| | Bert | 3.93e-9 | 0.4003 | 1.0 | - | - | - | - | - | - |
| | RoBERTa | 2.22e-7 | 0.1630 | 0.01383 | 1.0 | - | - | - | - | - |
| | Albert | 1.50e-8 | 0.7661 | 0.5451 | 0.4207 | 1.0 | - | - | - | - |
| T5 | RoBERTa-EE | 0.00103 | 3.82e-6 | 4.50e-7 | 1.98e-5 | 1.94e-6 | 1.0 | - | - | - |
| | RoBERTa-QA | 4.73e-9 | 0.2636 | 0.3299 | 0.4448 | 0.5182 | 1.88e-7 | 1.0 | - | - |
| | XLNet | 9.84e-9 | 0.2955 | 0.04332 | 0.02603 | 0.5797 | 9.44e-7 | 0.2931 | 1.0 | - |
| | Bert | 9.26e-9 | 0.1119 | 0.2184 | 0.4417 | 0.2108 | 2.22e-7 | 0.1858 | 0.03288 | 1.0 |

Table 18: p-values from Wilcoxon Sign-Ranked Test

## A.10 Qualitative Study of Bert-based Model

We include some examples of how our proposed system interact with user in Table 19 - 20. In those experiments, we use Bert as Transformer-based Language Model fine-tuned on Entity Extraction task (Table 19) and Question Answering task (Table 20).

| Turn | Conversation |
|------|--------------|
| **System setup:** | model: BERT, embedding: use, similarity function: cosine, schema: student performance |
| **Example No.** | 1 |
| **User:** | Predict the total number of school absences where the student's family size is greater than 3 for next year. |
| **agent:** | Thank you! It looks like the attribute of your goal prediction task is absences, is that correct? |
| **User:** | yes. |
| **agent:** | Thanks for confirming! Let's focus on the aggregator of your goal prediction task. It looks like for the aggregator you want to use TOTAL, is that correct? |
| **User:** | yeah that is correct. |
| **agent:** | Thanks for confirming!Let's focus on the filter of your goal prediction task. It looks like for the filter you want to use famsize, is that correct? |
| **User:** | yes. |
| **agent:** | Thanks for confirming!Let's focus on the filter operation of your goal prediction task. It looks like for the filter Operation you want to use NONE, is that correct? |
| **User:** | no. |
| **agent:** | Okay, so then is the filter operation of your goal prediction task should be NOT? |
| **User:** | no. |
| **agent:** | Okay, so then is the filter operation of your goal prediction task should be IS LESS THAN? |
| **User:** | no. |
| **agent:** | Okay, so then is the filter operation of your goal prediction task should be IS MORE THAN? |
| | Continued on next page |

**Table 19 – continued from previous page**

| Turn | Conversation |
|------|--------------|
| **User:** | yes. |
| **agent:** | Thanks for confirming!So this is the problem you are trying to solve: 
 Predict the attribute: absences 
 aggregator: total 
 filter: famsize 
 filter operation: more than . Is that correct? |
| **User:** | that seems all right. |
| **Example No.** | 2 |
| **System setup:** | model: BERT, embedding: use, similarity function: cosine, schema: online delivery |
| **User:** | predict the average number of calls made by delivery captain where the delivery persons ability is not good for the next month |
| **agent:** | Thank you! It looks like the attribute of your goal prediction task is Number of calls, is that correct? |
| **User:** | yes |
| **agent:** | Thanks for confirming!Let's focus on the aggregator of your goal prediction task. It looks like for the aggregator you want to use AVERAGE, is that correct? |
| **User:** | yes |
| **agent:** | Thanks for confirming!Let's focus on the filter of your goal prediction task. It looks like for the filter you want to use Delivery person ability, is that correct? |
| **User:** | yes. |
| **agent:** | Thanks for confirming!Let's focus on the filter operation of your goal prediction task. It looks like for the filter operation you want to use NONE, is that correct? |
| **User:** | no. |
| **agent:** | Okay, so then is the filter operation of your goal prediction task should be IS BETWEEN? |
| **User:** | still incorrect. Please check it again. |
| **agent:** | Okay, so then is the filter operation of your goal prediction task should be IS LESS THAN? |
| **User:** | nope |
| **agent:** | Okay, so then is the filter operation of your goal prediction task should be NOT? |
| **User:** | no wrong again. |
| **agent:** | Okay, so then is the filter operation of your goal prediction task should be IS EQUAL TO? |
| **User:** | yes |
| **agent:** | Thanks for confirming!So this is the problem you are trying to solve: 
 Predict the attribute: Number of calls 
 aggregator: AVERAGE 
 filter: Delivery person ability 
 filter operation: IS EQUAL TO |
| **Example No.** | 3 |
| **System setup:** | model: BERT, embedding: use, similarity function: cosine, schema: online delivery |
| **User:** | predict the total delay of delivery person getting assigned where the customers location on google map is not accurate for the next month |
| **agent:** | Thank you! It looks like the attribute of your goal prediction task is Delay of delivery person getting assigned, is that correct? |
| **User:** | yes |
| **agent:** | Thanks for confirming!Let's focus on the aggregator of your goal prediction task. It looks like for the aggregator you want to use TOTAL, is that correct? |
| **User:** | you are correct |
| **agent:** | Thanks for confirming!Let's focus on the filter of your goal prediction task. It looks like for the filter you want to use Google Maps Accuracy, is that correct? |
| **User:** | yes that is. |
| **agent:** | Thanks for confirming!Let's focus on the filter of your goal prediction task. It looks like for the filter you want to use Google Maps Accuracy, is that correct? |
| **User:** | yeah. |
| **agent:** | Thanks for confirming!Let's focus on the filter operation of your goal prediction task. It looks like for the filter operation you want to use NONE, is that correct? |

**Table 19 – continued from previous page**

| Turn | Conversation |
|---|---|
| **User:** | no. |
| **agent:** | Okay, so then is the filter operation of your goal prediction task should be IS EQUAL TO? |
| **User:** | no. |
| **agent:** | Okay, so then is the filter operation of your goal prediction task should be HAS? |
| **User:** | no. |
| **agent:** | Okay, so then is the filter operation of your goal prediction task should be IS LESS THAN? |
| **User:** | no. |
| **agent:** | Okay, so then is the filter operation of your goal prediction task should be IS GREATER THAN? |
| **User:** | no. |
| **agent:** | Okay, so then is the filter operation of your goal prediction task should be NOT? |
| **User:** | no. |
| **agent:** | Okay, so then is the filter operation of your goal prediction task should be NOT? |
| **User:** | yes that's correct. |
| **agent:** | Thanks for confirming!So this is the problem you are trying to solve: |
| | Predict the attribute: Delay of delivery person getting assigned |
| | aggregator: TOTAL |
| | filter: Google Maps Accuracy |
| | filter operation: NOT |
| **Example No.** | 4 |
| **System setup:** | model: BERT, embedding: use, similarity function: cosine, schema: flight delay |
| **User:** | Predict total arrival delay for each destination airports whose flights will start next Sunday |
| **agent:** | Thank you! It looks like the attribute of your goal prediction task is ARRIVAL DELAY, is that correct? |
| **User:** | that is correct |
| **agent:** | Thanks for confirming! Let's focus on the aggregator of your goal prediction task. |
| | It looks like for the aggregator you want to use TOTAL, is that correct? |
| **User:** | yes |
| **agent:** | Thanks for confirming!Let's focus on the filter of your goal prediction task. |
| | It looks like for the filter you want to use AIRLINE DELAY, is that correct? |
| **User:** | no i don't want any filter |
| **agent:** | filter of your goal prediction task is NONE? |
| **User:** | yes |
| **agent:** | Thanks for confirming!Let's focus on the filter operation of your goal prediction task. |
| | It looks like for the filter operation you want to use NOT, is that correct? |
| **User:** | no i don't want any filter |
| **agent:** | filter operation of your goal prediction task is NONE? |
| **User:** | yes that's correct |
| **agent:** | Thanks for confirming!So this is the problem you are trying to solve: |
| | Predict the attribute: ARRIVAL DELAY |
| | aggregator: TOTAL |
| | filter: NONE |
| | filter operation: NONE |

Table 19: Qualitative study of User Interaction with our proposed agent BERT as Transformer based Language Model which was fine-tuned on Entity extraction task.

| Turn | Conversation |
|---|---|
| **System setup:** | model: BERT, embedding: use, similarity function: cosine, schema: student performance |
| **Example No.** | 1 |
| **User:** | Predict the total number of school absences where the student's family size is greater than 3 for next year. |
| **agent:** | Thank you! It looks like the attribute of your goal prediction task is absences, is that correct? |

**Table 20 – continued from previous page**

| Turn | Conversation |
|---|---|
| **User:** | yes. |
| **agent:** | Thanks for confirming!Let's focus on the aggregator of your goal prediction task. It looks like for the aggregator you want to use TOTAL, is that correct? |
| **User:** | yeah that is correct. |
| **agent:** | Thanks for confirming! Let's focus on the filter of your goal prediction task. It looks like for the filter you want to use schoolsup, is that correct? |
| **User:** | no. |
| **agent:** | filter of your goal prediction task is G1, am I correct? |
| **User:** | no its family size |
| **agent:** | filter of your goal prediction task the famsize? |
| **User:** | yes. |
| **agent:** | Thanks for confirming!Let's focus on the filter operation of your goal prediction task. It looks like for the filter operation you want to use NOT, is that correct? |
| **User:** | no. |
| **agent:** | Okay, so then is the filter operation of your goal prediction task should be IS MORE THAN? |
| **User:** | yes. |
| **agent:** | Thanks for confirming!So this is the problem you are trying to solve: |
| | Predict the attribute: absences |
| | aggregator: total |
| | filter: famsize |
| | filter operation: more than . Is that correct? |
| **User:** | that seems all right. |
| **Example No.** | 2 |
| **System setup:** | model: BERT, embedding: use, similarity function: cosine, schema: online delivery |
| **User:** | predict the average number of calls made by delivery captain where the delivery persons ability is not good for the next month |
| **agent:** | Thank you! It looks like the attribute of your goal prediction task is Number of calls, is that correct? |
| **User:** | yes |
| **agent:** | Thanks for confirming!Let's focus on the aggregator of your goal prediction task. It looks like for the aggregator you want to use AVERAGE, is that correct? |
| **User:** | yes |
| **agent:** | Thanks for confirming!Let's focus on the filter of your goal prediction task. It looks like for the filter you want to use Delivery person ability, is that correct? |
| **User:** | yes. |
| **agent:** | Thanks for confirming!Let's focus on the filter operation of your goal prediction task. It looks like for the filter operation you want to use NOT, is that correct? |
| **User:** | no its should be equal |
| **agent:** | filter operation of your goal prediction task the IS EQUAL TO? |
| **User:** | yeah there you go. |
| **agent:** | Thanks for confirming!So this is the problem you are trying to solve: |
| | Predict the attribute: Number of calls |
| | aggregator: AVERAGE |
| | filter: Delivery person ability |
| | filter operation: IS EQUAL TO |
| **Example No.** | 3 |
| **System setup:** | model: BERT, embedding: use, similarity function: cosine, schema: online delivery |
| **User:** | predict the total delay of delivery person getting assigned where the customers location on google map is not accurate for the next month |
| **agent:** | Thank you! It looks like the attribute of your goal prediction task is Delay of delivery person getting assigned, is that correct? |
| **User:** | yes |
| **agent:** | Thanks for confirming! Let's focus on the aggregator of your goal prediction task. It looks like for the aggregator you want to use TOTAL, is that correct? |
| **User:** | you are correct |

**Table 20 – continued from previous page**

| Turn | Conversation |
|---|---|
| **agent:** | Thanks for confirming!Let's focus on the filter of your goal prediction task. |
| | It looks like for the filter you want to use Google Maps Accuracy, is that correct? |
| **User:** | yes that is. |
| **agent:** | Thanks for confirming!Let's focus on the filter operation of your goal prediction task. |
| | It looks like for the filter operation you want to use NOT, is that correct? |
| **User:** | yes that is correct. |
| **agent:** | Thanks for confirming!So this is the problem you are trying to solve: |
| | Predict the attribute: Delay of delivery person getting assigned |
| | aggregator: TOTAL |
| | filter: Google Maps Accuracy |
| | filter operation: NOT |
| **Example No.** | 4 |
| **System setup:** | model: BERT, embedding: use, similarity function: cosine, schema: flight delay |
| **User:** | Predict total arrival delay for each destination airports whose flights will start next Sunday |
| **agent:** | Thank you! It looks like the attribute of your goal prediction task is ARRIVAL DELAY, is that correct? |
| **User:** | that is correct |
| **agent:** | Thanks for confirming! Let's focus on the aggregator of your goal prediction task. |
| | It looks like for the aggregator you want to use TOTAL, is that correct? |
| **User:** | yes |
| **agent:** | Thanks for confirming!Let's focus on the filter of your goal prediction task. |
| | It looks like for the filter you want to use FLIGHT NUMBER, is that correct? |
| **User:** | no i don't want any filter |
| **agent:** | filter of your goal prediction task is NONE? |
| **User:** | yes |
| **agent:** | Thanks for confirming!Let's focus on the filter operation of your goal prediction task. |
| | It looks like for the filter operation you want to use NOT, is that correct? |
| **User:** | no i don't want any filter |
| **agent:** | filter operation of your goal prediction task is NOT, am I correct? |
| **User:** | no |
| **agent:** | Okay, so then is the filter operation of your goal prediction task should be HAS? |
| **User:** | no |
| **agent:** | filter operation of your goal prediction task the NONE? |
| **User:** | yes that's correct |
| **agent:** | Thanks for confirming!So this is the problem you are trying to solve: |
| | Predict the attribute: ARRIVAL DELAY |
| | aggregator: TOTAL |
| | filter: NONE |
| | filter operation: NONE |

Table 20: Qualitative study of User Interaction with our proposed agent BERT as Transformer based Language Model which was fine-tuned on Question Answering task.

## A.11 Qualitative study of Open-Domain Dialog Systems

The full qualitative study of ChatGPT on this ambitious task is presented in Table 21 - 24. In each of the table we have given different system instruction and see how the performance of ChatGPT varies with level of information provided to it before the prompt. From the examples we can see more informative system setup is more effective.

| Turn | Conversation |
|---|---|
| **System setup:** | You are a Machine Learning expert who will help a small business owner by formulating an interesting problem through conversation. You should perform a slot-filling task based on the conversation and use those slot information to formulate a time series forecasting problem for the user. The slots are aggregation operation, target attribute, filter attribute, and filter operation. you have to infer the slot-values from the conversations. |
| **Example No.** | 1 |
| **User:** 

 **chatGPT:** | Predict the total number of school absences where the student's family size is greater than 3 for next year. 
 Great! To start, I need some additional information to formulate a precise problem. Can you tell me more about the data you have regarding the school absence? |
| **User:** 
 **chatGPT:** | I do not have the data. Please use the above information for prediction. 
 Okay, I will need to identify the slots in the given information to formulate the problem precisely. 
 The Aggregation operation: 'total number' 
 Target attribute: 'school absence' 
 Filter attribute: 'family size' 
 Filter operation: 'greater than 3' |
| **User:** 
 **chatGPT:** | Thank you. 
 You're welcome! Feel free to let me know if you have any more questions or if there's anything else I can help you with. |
| **Example No.** | 2 |
| **User:** 

 **chatGPT:** | predict the average number of calls made by delivery captain where the delivery persons ability is not good for the next month 
 Sure, I can help you with that. To clarify, are you looking to predict the average number of calls made by delivery captains for the next month, but only when the delivery person's ability is poor? And could you please provide me with more information on how you define poor ability (i.e., what attributes are associated with poor ability)? |
| **User:** 
 **chatGPT:** | I do not have any more information 
 Okay, I can work with that. In that case, we can formulate the problem as a time series forecasting task where the target attribute is the average number of calls made by delivery captains, and the filter attribute is the delivery person's ability. Specifically, we want to predict the average number of calls made by delivery captains for the next month, but only when the delivery person's ability is poor. To do so, we will need to gather data on the average number of calls made by delivery captains for each month, as well as some measure of the delivery person's ability (i.e., a numerical score or rating). Once we have this data, we can apply a filter operation to only consider the data points where the delivery person's ability score is below a certain threshold (which we will determine based on the data). Finally, we can apply an aggregation operation to compute the average number of calls made by delivery captains for each month, after filtering for poor delivery person ability, and use this historical data to train a time series forecasting model that will predict the average number of calls for the next month, given the poor delivery person ability filter. |
| **User:** 
 **chatGPT:** | can you list all the slot-values here 
 Sure, here are the identified slot-values based on our conversation: 
 - Aggregation operation: Average 
 - Target attribute: Number of calls made by delivery captains 
 - Filter attribute: Delivery person's ability 
 - Filter operation: Poor ability |
| **User:** 
 **chatGPT:** | filter operation is incorrect. Please check it. 
 I apologize for the mistake. Can you please correct me? What would be the correct filter operation for the problem we discussed? |
| **User:** 
 **chatGPT:** | I do not have the correct answer. Read the original prompt. 
 You're right. I apologize for the confusion. Based on the prompt, the correct filter operation would be "delivery persons ability is not good". |
| **User:** 
 **chatGPT:** | still incorrect. Please check it again. 
 I'm sorry for the mistake. Let me verify this again for you. Could you please clarify what the right filter operation is? |
| **User:** | Hint: filter operation is conditional check operation. Now, check for the correcting the prompt |

**Table 21 – continued from previous page**

| Turn | Conversation |
|------|-------------|
| **chatGPT:** | Thank you for the hint. Based on the prompt, the correct filter operation for the problem would be a conditional check operation, specifically "where the delivery persons ability is not good". This means that we would be filtering data to only consider cases where the delivery person's ability is below a certain threshold, which we would need to determine based on the data. |
| **Example No.** | 3 |
| **User:**

**chatGPT:** | predict the total delay of delivery person getting assigned where the customers location on google map is not accurate for the next month
Sure, I can help you with that. Can you please tell me more about your business and what kind of industry you are in? Also, how do you currently manage your deliveries? |
| **User:**
**chatGPT:** | i do not have any more information.
Oh, I see. Let me clarify few things, when you say "total delay of delivery person getting assigned", do you mean
the time taken to assign a delivery person to a customer's order? And by "where the customer's location on Google map is not
accurate", do you mean the location is not mapped correctly or that it is unclear where exactly the location is? |
| **User:**
**chatGPT:** | can you list the slot-values here
Sure, based on our conversation, I have identified the following slots:
- Aggregation operation: Total
- Target attribute: delay of delivery person getting assigned
- Filter attribute: customer's location on Google map accuracy
- Filter operation: inaccurate
So the complete problem statement would be:
"Forecast the total delay of delivery person getting assigned, where the customer's location on Google map is inaccurate for the next month." |
| **User:**
**chatGPT:** | Thank you.
You're welcome! Feel free to let me know if you have any more questions or if there's anything else I can help you with. |
| **Example No.** | 4 |
| **User:**
**chatGPT:** | Predict total arrival delay for each destination airports whose flights will start next Sunday
Sure, we can work on that. Before we proceed, can you provide me some more information like which airline's flights are we considering and which geographical region of the world? Also, can you tell me are we interested in predicting delays for a specific time period, such as a single day or a week, or a longer period perhaps multiple weeks or month? |
| **User:**
**chatGPT:** | I do not have any additional information
Alright, let me see if I understood it correctly. We want to predict the total arrival delay for each destination airport for flights that will start next Sunday. But we do not have any information about the airline or the geographical region. And we are interested in predicting delays for a specific day, Sunday. Is that correct? |
| **User:**
**chatGPT:** | yes
Great. So based on this slot-filling conversation, we can formulate the following time series forecasting problem:
Problem Statement: Predict the total arrival delay for each destination airport for flights that will start next Sunday. Slots:
- Aggregation Operation: Total
- Target Attribute: Arrival Delay
- Filter Attribute: Destination Airport
- Filter Operation: Next Sunday's flights |
| **User:**
**chatGPT:** | filter attribute and filter operation is incorrect
I apologize for the mistake. Can you please tell me what is the correct filter attribute and filter operation? With that information, I will update the problem description for you. |
| **User:**
**chatGPT:** | no. i didn't have any information
I see. In that case, we can formulate the problem without the filter attribute and filter operation slots as follows: |

Continued on next page

**Table 21 – continued from previous page**

| Turn | Conversation |
|------|--------------|
|  | Problem Statement: Predict the total arrival delay for each destination airport for flights that will start next Sunday. Slots: 
 - Aggregation Operation: Total 
 - Target Attribute: Arrival Delay 
 Time Period: A specific day, Sunday |
| **User:**: | thank you very much |

Table 21: Qualitative study of User Interaction with ChatGPT using the same system setup described in the beginning.

| Turn | Conversation |
|---|---|
| **System setup:** | You are a personal data scientist who will help a small business owner by formulating an interesting problem through conversation. You should perform a slot-filling task based on the conversation and use those slot information to formulate a time series forecasting problem for the user. The slots are aggregation operation, target attribute, filter attribute, and filter operation. you have to infer the slot-values from the conversations. |
| **Example No.** | 1 |
| **User:** | Predict the total number of school absences where the student's family size is greater than 3 for next year. |
| **chatGPT:** | Thank you for providing the time series forecasting problem regarding school absence. Now, I need some more information to infer the slot-values from the given conversation. Can you please provide me with the conversation between you and the small business owner that you had regarding this topic, so that I can identify the slot-values and help you with the task? |
| **User:** **chatGPT:** | I do not have any other information between me and small business owner.
Alright, based on the forecasting problem that you have provided, I can infer the following slot-values:
- Aggregation operation: "sum" (since we want to predict the total number of school absences)
- Target attribute: "school absence" (since this is what we want to predict)
- Filter attribute: "family size" (since we want to consider only those instances where the family size is greater than 3)
- Filter operation: ">" (since we want to filter those instances where the family size is greater than 3)
Based on the inferred slot-values, the time series forecasting problem would be:
Forecast the sum of school absences where the family size is greater than 3, for the next year. |
| **User:** **chatGPT:** | That is correct. Thank you.
You're welcome! Feel free to let me know if you have any more questions. |
| **Example No.** | 2 |
| **User:** **chatGPT:** | Predict the maximum first period grade of students where student has partaken in extra-curricular activities
I'm sorry, but I cannot perform this task as you have provided a time series forecasting problem without any conversation or slot information. To perform the task, please provide the necessary details. |
| **User:** **chatGPT:** | I have no new information regarding the problem statement. Please perform the task using the provided information
I'm sorry, but to perform a time series forecasting problem, we need a time series dataset along with the slots that specify the aggregation operation, target attribute, filter attribute, and filter operation for data preprocessing. Without this information, it is difficult to formulate a suitable time series forecasting model. Can you please provide more information about the problem statement? |
| **User:** **chatGPT:** | i do not have more information. please use the original prompt to find the solution.
I apologize for the confusion, but the original prompt does not provide any conversation or slot information to perform the slot filling task. The prompt only mentions a time series forecasting problem without any context. Can you please provide more details or context about the problem statement so that I can assist you better? |
| **User:** | ok. |
| **Example No.** | 3 |
| **User:** **chatGPT:** | predict the average number of calls made by delivery captain where the delivery persons ability is not good for the next month.
Sure, can you provide some more details to help me understand the context better? What kind of business are you running, and what is the role of the delivery captain and delivery persons? Also, can you provide the conversation that led to this query to help me fill in the slots? |
| **User:** **chatGPT:** | i do not have any other information
Alright, based on the limited information available, let me try to fill in the slots as follows:
- Aggregation operation: "average"
- Target attribute: "number of calls made by delivery captain"
- Filter attribute: "delivery person's ability"
- Filter operation: "not good" |

**Table 22 – continued from previous page**

| Turn | Conversation |
|------|-------------|
| **Example No.** | 4 |
| **User:** | predict the total delay of delivery person getting assigned where the customers location on google map is not accurate for the next month |
| **chatGPT:** | Let me clarify the slots based on the statement you provided. |
| | Aggregation operation: Summation |
| | Target attribute: Delay of delivery person getting assigned |
| | Filter attribute: Accuracy of customer location on Google Maps |
| | Filter operation: Inaccurate |
| | So, the slot values are as follows: |
| | Aggregation operation: Summation |
| | Target attribute: delay of delivery person getting assigned |
| | Filter attribute: accuracy of customer location on Google Maps |
| | Filter operation: inaccurate |
| | To formulate a time series forecasting problem, we need historical data about the delay of delivery person getting assigned for inaccurate locations on Google Maps. Do you have any data that we can use for this purpose |
| **User:** | you got it right |
| **Example No.** | 5 |
| **User:** | Predict total arrival delay for each destination airports whose flights will start next Sunday |
| **chatGPT:** | I'm sorry, but I think there's some misunderstanding. I'm supposed to perform a slot filling task based on conversation with a small business owner and formulate a time series forecasting problem using those slot information. Could you please provide me with some information about the conversation? |
| **User:** | I do not have more information |
| **chatGPT:** | No problem, I'll give an example conversation to illustrate how slot filling can be performed: |
| | Small Business Owner: Hi, can you help me analyze my sales data for the past year? |
| | Data Scientist: Sure, what specific aspect do you want me to focus on? |
| | Small Business Owner: I want to know how the sales have been for our top product. |
| | Data Scientist: Alright, what is the name of your top product? |
| | Small Business Owner: It's called Widget A. |
| | Data Scientist: Got it. Do you have any specific time period you want me to analyze? |
| | Small Business Owner: Yes, I want to see the sales trend for Widget A for the past 12 months. |
| | From this conversation, we can infer the following slot information: |
| | - Aggregation operation: None (no aggregation requested) |
| | - Target attribute: Sales |
| | - Filter attribute: Product Name |
| | - Filter operation: Equal to |
| | Using this information, we can formulate a time series forecasting problem as follows: |
| | "Forecast the sales trend for Widget A for the next 12 months, given the sales data from the previous 12 months." This problem involves time series forecasting, where we will use historical sales data to predict future sales for a specific product. |
| **User:** | the above slot filling is incorrect. please redo the slot-filling task |
| **chatGPT:** | My apologies for the confusion earlier. Could you please provide me with a conversation between you and the small business owner so that I can perform the slot filling task? |

**Table 22 – continued from previous page**

| Turn | Conversation |
|---|---|
| **User:** | I do not have any such information |
| **chatGPT:** | Not a problem. I'll provide an example conversation and perform the slot filling task: |
| | Small Business Owner: Hi, can you help me with my inventory management? |
| | Data Scientist: Of course, what specifically do you need help with? |
| | Small Business Owner: I need to know which products are selling the most and which ones are not doing so well. |
| | Data Scientist: Understood. For which time period do you want me to analyze the sales data? |
| | Small Business Owner: Please analyze the data for the past 6 months. |
| | From this conversation, we can infer the following slot information: |
| | - Aggregation operation: Maximum count (to identify the top-selling products) |
| | - Target attribute: Sales |
| | - Filter attribute: Time |
| | - Filter operation: Less than or equal to |
| | Using this information, we can formulate a time series forecasting problem as follows: |
| | "Forecast the sales trend for the top-selling products and the non-performing products for the next 3 months, given the sales data for the past 6 months." This problem involves time series forecasting, where we will use historical sales data to predict the sales trend for the top-selling and non-performing products. |
| **User:** | the slot-filling is incorrect. |
| | the same iteration continues until user quits. |

Table 22: Qualitative study of User Interaction with ChatGPT using the same
system setup described in the beginning.

| Turn | Conversation |
|---|---|
| **System setup:** | You are a personal data scientist who will help a small business owner by formulating an interesting problem through conversation. You should perform a slot filling task based on the conversation and use those slot information to formulate a time series forecasting problem for the user. The slots are aggregation operation, target attribute, filter attribute and filter operation. you have to infer the slot-values from the conversations. The value of the slots are: (predefined values for each possible slots are given here.) |
| **Example No.** | 1 |
| **User:** 
 **chatGPT:** | Predict total arrival delay for each destination airports whose flights will start next Sunday 
 Alright. To clarify, you are looking for a time series forecasting problem to predict the total arrival delay for each destination airport for flights that will begin on next Sunday. Is that correct? |
| **User:** 
 **chatGPT:** | yes. 
 Great! So to formulate a time series forecasting problem, we'll need to define the time interval we're looking at. Based on the given slots, we can use the date or day of the week as the time variable. 
 For this problem, we'll use the date of flight departure as the time variable, and since we're only interested in flights departing next Sunday, we can filter the dataset by the "date" attribute using the "equal" filter operation. 
 We'll also use the "arrival delay" attribute as the target variable, and the "tail number" and "destination airport" attributes as our categorical variables. To aggregate the data, we'll use the "total" aggregation operation, since we're interested in the total arrival delay for each destination airport. 
 So our final problem statement is: Given flight data for all flights departing next Sunday, predict the total arrival delay for each destination airport. The categorical variables used for prediction are "tail number" and "destination airport". |
| **User:** | correct. |
| **Example No.** | 2 |
| **User:** 


 **chatGPT:** | predict the average air system delay for flights which will start from Atlanta International Airport which will have scheduled departure hour after 4 PM is central time and will start within next week 
 To clarify, do you want the predicted average air system delay for all flights meeting the criteria you specified, or for a specific flight or set of flights? Additionally, do you have access to historical data on flights from Atlanta International Airport to use for the time series forecasting? |
| **User:** 
 **chatGPT:** | i do not have access to historical data 
 Without access to historical data, it would not be possible to perform a time series forecasting of the average air system delay for flights meeting the criteria you specified. However, based on the information provided, we can still generate an estimate of the average air system delay for those flights. 
 To do so, we would need to gather data on recent flights departing from Atlanta International Airport during the specified time range and calculate the average air system delay for those flights. One way to gather this data would be to use an API such as the FlightStats API or the Aviationstack API to retrieve real-time flight data for flights departing from Atlanta International Airport. 
 Once the data has been gathered, we would then need to filter the data based on the specified criteria (scheduled departure hour after 4 PM in central time, departing within the next week). We could then calculate the average air system delay for the filtered data to obtain our estimate. 
 The aggregation operation for this task would be 'average', the target attribute would be 'air system delay', and the filter attributes would be 'scheduled departure hour' and 'date'. The filter operation would be 'greater than' or 'after' for the 'scheduled departure hour' attribute and 'between' or 'about' for the 'date' attribute. |
| **User:** 
 **chatGPT:** | correct. Thank you. 
 You're welcome! If you have any further questions or need additional assistance, feel free to ask. |
| **Example No.** | 3 |

**Table 23 – continued from previous page**

| Turn | Conversation |
|---|---|
| **User:** | Predict the total number of school absence where student's family size is greater than 3 for next year |
| **chatGPT:** | To formulate a time series forecasting problem, we would need historical data of the number of school absences for the past few years. Assuming that we have that data, we can use the following slot information to filter the relevant data: 
 - Aggregation operation: total 
 - Target attribute: absences 
 - Filter attribute: famsize 
 - Filter operation: greater than 
 - Filter value: 3 
 Using this information, we can filter the historical data to only include records where the student's family size is greater than 3 and calculate the total number of school absences for each year. We can then use this data to create a time series model to forecast the total number of school absences for the next year. The model can use various time series forecasting techniques such as ARIMA or exponential smoothing to make predictions based on the patterns observed in the historical data. |
| **User:** | there is no filter value |
| **chatGPT:** | Oh, I apologize for the oversight. To complete the slot filling task, we will need a filter value for the "family size" attribute. Let's assume that the filter value is "4" indicating students with a family size greater than 3 (i.e., students who have 4 or more family members in their household). So, the complete slot information would be: 
 - Aggregation operation: total 
 - Target attribute: absences 
 - Filter attribute: famsize 
 - Filter operation: greater than 
 - Filter value: 3 
 We can use this slot information to filter the historical data and make predictions for the total number of school absences for students with family size greater than 3 for the next year. |
| **User:** | we did not asked for filter value |
| **chatGPT:** | I apologize for the confusion. Given the updated slot information, the corresponding time series forecasting problem would be to predict the total number of school absences for students with a family size greater than 3 for the next year. To do this, we would extract the relevant information from the historical data, apply any necessary data cleaning and preprocessing steps, then use a time series forecasting model to project the expected number of school absences for the next year. The model could be built using techniques such as ARIMA, exponential smoothing or machine learning algorithms like LSTM or GRU, depending on the data and the specific requirements of the problem. |
| **User:** | can you now show the complete slot information |
| **chatGPT:** | Sure, here is the complete slot information: 
 - Aggregation operation: total 
 - Target attribute: absences 
 - Filter attribute: famsize 
 - Filter operation: greater than 
 - Filter value: 3 
 Using this slot information, we would extract the relevant data for students with a family size greater than 3, apply any necessary preprocessing such as missing data imputation and feature scaling, and build a time series forecasting model to predict the total number of school absences for the next year. The model would be trained on the historical data and evaluated based on its performance on a holdout dataset. If the model's performance is satisfactory, it could be used to make predictions for future years as well. |

**Table 23 – continued from previous page**

| Turn | Conversation |
|---|---|
| **User:** | can you remove filter value as i did not ask for it. |
| **chatGPT:** | Apologies for the confusion. Here's the updated slot information without the filter values: |
| | - Aggregation operation: total |
| | - Target attribute: absences |
| | - Filter attribute: famsize |
| | - Filter operation: greater than |
| | Given this slot information, we would filter the historical data to only include records where the student's family size is greater than 3 without any specific value of family size. We can then use the filtered historical data to train a time series forecasting model to predict the total number of school absences for next year for students with family size greater than 3. The model can be trained using various time series forecasting techniques such as ARIMA or exponential smoothing to make predictions based on the patterns observed in the historical data. |
| **User:** | Thank you. |
| **chatGPT:** | You're welcome! If you have any more questions or if there's anything else I can help you with, feel free to ask. |
| **Example No.** | 4 |
| **User:** | Predict the maximum first period grade of students where student has partaken in extra-curricular activities |
| **chatGPT:** | So, the aggregation operation is 'maximum', the target attribute is 'g1', the filter attribute is 'activities' and the filter operation is 'has'. Therefore, we need to collect the data of those students who have partaken in extra-curricular activities, and then predict the maximum first period grade among them using a time series forecasting model. |
| **User:** | That is correct. Thank you. |
| **chatGPT:** | You're welcome! If you have any further questions or need any help in the future, feel free to ask. |
| **Example No.** | 4 |
| **User:** | predict the average number of calls made by delivery captain where the delivery persons ability is not good for the next month |
| **chatGPT:** | To formulate a time series forecasting problem, we will need a time series data that has information about the average number of calls made by delivery captain and also if the delivery person's ability was not good for that particular order delivery. |
| | Assuming we have the required data, we can use the following slot information to formulate the problem: |
| | Aggregation operation: average |
| | Target attribute: number of calls |
| | Filter attribute: delivery person ability |
| | Filter operation: not good |
| | Using this information, we can frame the problem statement as: |
| | "Predict the average number of calls made by delivery captain, where the delivery person's ability was not good, for the next month." |
| | The next step would be to choose an appropriate time series forecasting model (such as ARIMA, exponential smoothing, Prophet, etc.) and train it on the available historical data. Once the model is trained, we can use it to forecast the average number of calls for the next month. |
| **User:** | That is correct. Thank you. |
| **chatGPT:** | You're welcome! If you have any further questions or need any help in the future, feel free to ask. |
| **Example No.** | 5 |

Continued on next page

**Table 23 – continued from previous page**

| Turn | Conversation |
|---|---|
| **User:** | predict the total delay of delivery person getting assigned where the customers location on google map is not accurate for the next month |
| **chatGPT:** | To formulate a time series forecasting problem for this scenario, we can define the following slots: |
| | - Aggregation operation: Total |
| | - Target attribute: Delay of delivery person getting assigned |
| | - Filter attribute: Google maps accuracy |
| | - Filter operation: Not, Equal to |
| | This means we want to predict the total delay of delivery person getting assigned for the next month, where the customer's location on Google maps is not accurate. |
| | To solve this problem we need historical data on: |
| | - Delay of delivery person getting assigned |
| | - Google maps accuracy |
| | - Time |
| | We also need external data (since we are predicting future delays): |
| | - Forecasted customer demand (we can obtain an estimate of the number of orders expected for the next month) |
| | Once we have this data, we can use various time-series models (such as ARIMA or SARIMA) to predict the total delay of delivery person getting assigned for the next month based on historical data and forecasted customer demand. |
| **User:** | can you write the code for above prompt |
| **chatGPT:** | Unfortunately, I cannot provide a code implementation for the prompt you mentioned without additional information such as the specific dataset, programming language, and framework to be used. |
| | However, I can provide a general outline of the steps involved in implementing the solution: |
| | 1. Collect historical data for the relevant attributes (Delay of delivery person getting assigned, Google maps accuracy, and time). |
| | 2. Clean and preprocess the data (e.g., remove duplicates and missing values, convert data types). |
| | 3. Use a time-series model (such as ARIMA or SARIMA) to train on the historical data. |
| | 4. Use the trained model to generate a forecast for the next month's total delay of delivery person getting assigned based on forecasted customer demand. |
| | 5. Evaluate the accuracy of the model using performance metrics such as the mean absolute error or root mean squared error. |
| | 6. Deploy the model to make real-time predictions for future delays based on new data. |
| | Please note that this is a general outline, and the specific implementation may vary depending on the data and the requirements of the project. |
| **User:** | Thank you for your help. |
| **chatGPT:** | You're welcome! If you have any further questions or need any help in the future, feel free to ask. |

Table 23: Qualitative study of User Interaction with ChatGPT using the same system setup described in the beginning.

| Turn | Conversation |
|------|--------------|
| **System setup:** | You are a Machine Learning expert who will help a small business owner by formulating an interesting problem through conversation. You should perform a slot filling task based on the conversation and use those slot information to formulate a time series forecasting problem for the user. The slots are aggregation operation, target attribute, filter attribute and filter operation. you have to infer the slot-values from the conversations. The value of the slots are: (predefined values for each possible slots are given here.) |
| **Example No.** | 1 |
| **User:** | Predict the total number of school absences where the student's family size is greater than 3 for next year. |
| **chatGPT:** | To formulate a time series forecasting problem based on the given information, we need to identify the relevant slots. The aggregation operation is 'total', the target attribute is 'school absence', and the filter attributes are 'family size' and 'year'. The filter operation is 'greater than'. However, we don't have any historical data for 'school absence', so we cannot create a time series forecasting problem.
Is there any other information regarding the data available? |
| **User:**
**chatGPT:** | No. I do not have any additional information.
In that case, we cannot formulate a time series forecasting problem using the given information. We would need historical data on school absence and family size in order to create a time series forecasting model. Without historical data, we cannot make predictions about the total number of school absences for next year.
However, we could still use the given information to formulate other types of predictive models. For example, we could create a classification model to predict whether a student will be absent from school based on their family size and other relevant factors. Alternatively, we could create a regression model to predict the number of school absences for a particular student based on their family size and other factors. |
| **User:**
**chatGPT:** | please reinitiate the process from first message.
Sure, let's start from the beginning. Hello! How can I assist you today? |
| **User:** | Predict the total number of school absence where student's family size is greater than 3 for next year |
| **chatGPT:** | To predict the total number of school absences where a student's family size is greater than 3 for next year, we need to formulate a time series forecasting problem. In order to do that, we need to identify the relevant slots based on the given information.
The aggregation operation is 'total', the target attribute is 'school absence', and the filter attributes are 'family size' and 'year'. The filter operation is 'greater than'. However, we don't have any historical data for 'school absence', so we cannot create a time series forecasting problem.
Do you have any historical data for school absences and family size for previous years? If yes, please provide the data. |
| **User:**
**chatGPT:** | what other data you required to create time series forecasting
We would need historical data on school absences and family size in order to create a time series forecasting model. Here are some additional pieces of information which would be useful:
- The time period for which the historical data is available (e.g. monthly, quarterly, annually)
- The range of dates/times covered by the data
- The format in which the data is available (e.g. CSV, Excel file, SQL database)
- Any other relevant data that might have an impact on school absences, such as average income of families, student demographics, or school policies and programs related to attendance.
With this information, we can use time series forecasting methods such as ARIMA or exponential smoothing to create a model that can predict future school absences based on historical trends and factors that impact attendance. |
| **User:**
**chatGPT:** | 
It seems like you didn't provide any prompt for me to assist you with. Is there anything related to machine learning or artificial intelligence that you would like me to help you with? |

**Table 24 – continued from previous page**

| Turn | Conversation |
|------|-------------|
| **User:** | we know that the prediction task is for next year and family size should be greater than 3. That the only data we have |
| **chatGPT:** | In that case, without any historical data, it is not possible to formulate a reliable time series forecasting model to predict the total number of school absences for next year where the student's family size is greater than 3. |
| | However, we could still use other types of predictive models to make some estimates based on the given information. For example, we might find a data set from another source, such as a government agency or educational institution, that provides information on school absences for students with families of varying sizes. We could then use that data to train a classification or regression model to predict school absences for students with families of size greater than 3. |
| | Alternatively, we could survey families with children in the relevant school district to gather information on family size and anticipated school absences for next year. We could use this survey data to train a predictive model such as logistic regression or decision tree regression to make predictions based on family size and other relevant factors. |
| **User:** | okay. thank you |
| **chatGPT:** | You're welcome. If you have any other questions related to machine learning or artificial intelligence, don't hesitate to ask! |
| **Example No.** | 2 |
| **User:** | Predict the maximum first period grade of students where student has partaken in extra-curricular activities |
| **chatGPT:** | I'm sorry, but the given problem statement and list of slots are related to online food delivery services. Can you please provide me with more information about the business you are referring to and the data available for analysis? |
| **User:** | The problem is to predict the first period grade students who has taken part in extra-curricular activities. |
| **chatGPT:** | Great! To help me formulate the problem, could you please provide me with the following details through slot filling - |
| | 1. What is the aggregation operation you are interested in? Do you want to predict the average, total, mean, max or minimum of the first period grade of students who have taken part in extra-curricular activities? |
| | 2. What is the target attribute you want to predict? Is it the first period grade? |
| | 3. Do you want to apply any filter on the data? If yes, what is the filter attribute and operation you want to use? For example, do you want to predict the first period grade for a specific age group of students or for students who have taken part in specific extra-curricular activities? |
| | 4. Do you have any additional information which can help in formulating the problem such as the time period the data was collected from or any other attributes about the students? |
| **User:** | Quits. |
| **Example No.** | 3 |
| **User:** | predict the average number of calls made by delivery captain where the delivery persons ability is not good for the next month |
| **chatGPT:** | Sure, I can help you with that. To clarify, the problem you are trying to solve is forecasting the average number of calls made by a delivery captain in a month when the delivery person's ability is not good. Can you please provide me with the filter attribute and filter operation to narrow down the scope of the problem? |

**Table 24 – continued from previous page**

| Turn | Conversation |
|---|---|
| **User:** | i do not have more data |
| **chatGPT:** | Alright. In that case, let's formulate the problem like this: |
| | Problem: Forecast the average number of calls made by delivery captains in a month when the delivery person's ability is not good. |
| | To do this, we will take a time series approach and use the historical data of number of calls made by delivery captains in the past months as our time series data. We will focus on the following slots: |
| | - Aggregation operation: average |
| | - Target attribute: number of calls made by delivery captain |
| | - Filter attribute: delivery person ability |
| | - Filter operation: not good |
| | We will gather data from the past 12 months, and use it to build a time series forecasting model that predicts the future average number of calls in a month where the delivery person's ability is not good. |
| **User:** | okay. thank you |
| **chatGPT:** | You're welcome. If you have any other questions related to machine learning or artificial intelligence, don't hesitate to ask! |
| **Example No.** | 4 |
| **User:** | predict the total delay of delivery person getting assigned where the customers location on google map is not accurate for the next month |
| **chatGPT:** | Sure, can you provide me with some information to help me understand the problem better? |
| | What is the industry or sector of the business? |
| | Also, how does the business currently track the delay of delivery person getting assigned? And do they have historical data on the delay of delivery person getting assigned in situations where a customer's location on google map is not accurate? |
| **User:** | i do not any other information |
| **chatGPT:** | Alright, based on the given slots, we can formulate the problem as follows: |
| | Problem Statement: Forecast the total delay of delivery person getting assigned in minutes for the next month, given that the customer's location on google map is not accurate. |
| | Aggregation Operation: Total |
| | Target Attribute: Delay of Delivery Person Getting Assigned |
| | Filter Attribute: Google Maps Accuracy |
| | Filter Operation: Not |
| | Time Series: Monthly |
| | Based on the available information, we have filtered out the cases where the customer's location on google map is not accurate. We will use historical data on the delay of delivery person getting assigned while considering only the cases where the customer's location on google map is not accurate. We will then use this information to forecast the total delay of delivery person getting assigned for the next month. |
| **User:** | remove time series slot from above prompt. I did not asked for that. |
| **chatGPT:** | Sure, here is the updated prompt without the time series slot: |
| | Problem Statement: Forecast the total delay of delivery person getting assigned in minutes, given that the customer's location on google map is not accurate. |
| | Aggregation Operation: Total |
| | Target Attribute: Delay of Delivery Person Getting Assigned |
| | Filter Attribute: Google Maps Accuracy |
| | Filter Operation: Not |
| | Using historical data of delay of delivery person getting assigned in cases where customer's location on google map was not accurate, we can predict the total delay of delivery person getting assigned in future. |
| **User:** | that is correct. Thanks! |
| **chatGPT:** | You're welcome. If you have any other questions related to machine learning or artificial intelligence, don't hesitate to ask! |
| **Example No.** | 5 |

**Table 24 – continued from previous page**

| Turn | Conversation |
|---|---|
| **User:** | Predict total arrival delay for each destination airports whose flights will start next Sunday |
| **chatGPT:** | Great! To help me understand your problem better, can we first narrow down the scope by fixing the aggregation operation, filter attribute and filter operation? It will be easier to formulate the problem. |
| **User:** | can you fill the slots based on the prompt provided |
| **chatGPT:** | Sure! So, based on the prompt, the 'aggregation operation' is 'total', the 'filter attribute' is 'day of week', and the 'filter operation' is 'equal'. Since you want to predict total arrival delay for each destination airport whose flights will start next Sunday, then the 'target attribute' should be 'arrival delay' and the 'filter value' should be 'Sunday'. Is that correct? |
| **User:** | yes. |
| **chatGPT:** | Great! Now that we have all the necessary slots filled, we can formulate the problem as follows: Problem Statement: Forecast the total arrival delay of flights for each destination airport, whose flights are scheduled for next Sunday. Aggregation Operation: Total Target Attribute: Arrival Delay Filter Attribute: Day of week Filter Value: Sunday Does this make sense? |
| **User:** | i do not need filter value, instead, i need filter operation. |
| **chatGPT:** | Sorry for the confusion. As per the prompt, we can set 'filter operation' to 'equal' since we want to predict arrival delay at each destination airport only for flights scheduled on Sunday. So, the complete problem statement is: Problem Statement: Forecast the total arrival delay of flights for each destination airport, whose flights are scheduled for next Sunday. Aggregation Operation: Total Target Attribute: Arrival Delay Filter Attribute: Day of Week Filter Operation: Equal |
| **User:** | correct. |

Table 24: Qualitative study of User Interaction with ChatGPT using the same system setup described in the beginning.

