# OpenReview forum: "Introducing "Forecast Utterance" for Conversational Data Science"
_TMLR — Accepted by TMLR_

### Review · Reviewer_7NQw · 2023-10-17

**Summary Of Contributions:**

The authors present the problem of forecasting by LLMs. They define a forecast by a few slots that one may fill (e.g., target attribute).
They provide a method to fill each of the slots given the human's input, those are based on Question answering or entity extraction and on simpler heuristics. The paper also annotates and validates about a thousand examples of sentences and filled slots for evaluation. The paper also creates augmentation data

**Audience:**

Yes

**Claims And Evidence:**

Yes

**Requested Changes:**

# minor
Unnecessary ", through extensive manual effort"
Use citep note \citet when the names are not part of the sentence. "the CoNLL-2003 format Tjong Kim Sang & De Meulder (2003) (for EE) and SQuAD format Rajpurkar
et al. (2016) (for QA)."

**Strengths And Weaknesses:**

The process is extensive, providing the full context for the problem, definition, data, evaluation, training data augmentation and a first baseline.

# Weaknesses
The intro and motivation are very well, but the actual proposal could be a bit more concrete at the beginning. The first figure doesn't really help, as it stays in the level of arbitrary letters and is not intuitive. It would help readers that skim or do not reach all the way in depth, and even just those that might want easier life while going through the rest of the paper if you provide more intuitive explanation of what you do (with short examples) as opposed to the overall idea of forecasting. Also a scheme of the process (user-> slot-> answer) might help. For example while reading I was waiting for the part where you explain how a language model predicts arbitrary sequences with you method, which is related but not exactly what you do.

---

> ### Author Response · Authors · 2023-10-30
> **Addressing Introduction Refinement, Figure 1 Revision, and Citation Format Update**
>
> Thank you for the invaluable feedback. Below is our plan to address the identified concerns regarding the introduction, Figure 1, and citation format.
> ## Introduction Refinement:
> In light of the feedback, we will revise the initial sections of the manuscript to provide a clearer and more concrete description of our method's objectives and novelty.
> ## Improvement of Figure 1:
> Following the reviewer's observation regarding the lack of intuitiveness in Figure 1, we plan to replace it with a schematic diagram. This diagram will visually represent the process from user input to slot filling to answer generation (user -> slot -> answer), aiming to provide a more intuitive explanation of our method. By incorporating illustrative examples within this schematic diagram, we intend to facilitate a more straightforward comprehension of the core concepts for all readers.
> ## Citation Format:
> We will update the citations in the manuscript to adhere to the requested format, utilizing the \citet and \citep commands as instructed.
>
>
> We are open to further suggestions to improve our manuscript and look forward to the opportunity to enhance its clarity and comprehensiveness.

---

### Review · Reviewer_8Xry · 2023-10-30

**Summary Of Contributions:**

The paper presents a novel approach to facilitate time series forecasting for non-experts through natural language conversations with an AI agent. The authors introduce the concept of Forecast Utterance, which captures users' prediction goals in natural language. They frame the task of understanding these utterances as an unsupervised slot-filling problem and propose two zero-shot methods, Entity Extraction (EE) and Question-Answering (QA), for solving it. The paper also presents three benchmark datasets for future research in this area and provides case studies to demonstrate the feasibility and effectiveness of the proposed techniques.

**Audience:**

Yes

**Claims And Evidence:**

Yes

**Requested Changes:**

Please refer to my concerns above.

**Strengths And Weaknesses:**

Strengths:

- The paper addresses a significant problem of making time series forecasting accessible to non-experts, which can have a substantial impact on various industries and businesses.
- The introduction of Forecast Utterance is a promising step towards making conversational data science more intuitive and user-friendly.
- The unsupervised slot-filling problem formulation and the two zero-shot approaches (EE and QA) are novel and well-suited to the presented problem.
- The creation of three benchmark datasets demonstrates a commitment to promoting future research in this area. And the case studies provide evidence of the feasibility and effectiveness of the proposed methods on real-world datasets.

Weaknesses:
- The paper could benefit from a more in-depth comparison of the two proposed zero-shot methods (EE and QA), discussing their relative strengths and weaknesses and the scenarios in which one might be preferred over the other.
- The unsupervised nature of the proposed methods may lead to potential issues in terms of accuracy and robustness. The paper should discuss possible limitations and provide suggestions for future work to address them.
- A more comprehensive evaluation, including quantitative metrics, can help assess the performance of the proposed methods and compare them with existing approaches.
- The paper does not address the potential challenges in handling ambiguous or complex user utterances that may require additional clarification or follow-up questions.

---

### Review · Reviewer_rYGP · 2023-12-23

**Summary Of Contributions:**

The paper presents a technique to do slot-filling for the task of ML problem formulation in the conversational data science domain. Specifically, the paper focuses on what kind of forecasting (i.e. regression but for values outside of the dataset's ranges), the task of identifying the relevant target variables, any filter operations needed on the user-provided data. The paper does this by first synthesizing a dataset of possible forecasting queries that can be made against the user-provided data leveraging it's schema, by using hand-crafted heuristic rules as well as employing a T5-based model. This synthetic data is then used to fine-tune several BERT-based model variants for the task of finding relevant segments within a query that can point to the target variable. The paper evaluates this technique on 3 Kaggle datasets.

**Audience:**

Yes

**Claims And Evidence:**

No

**Requested Changes:**

* Intro states “...each user may have unique data sets and datascience needs, it is unrealistic to assume that any training data is available to pre-train these conversational agents, making the supervised learning paradigm impractical.” Are there any stats or references that show whether user data science needs are diverse or not? (In other words, are we sure the most of the user needs are not satisfied by a small subset of techniques?)
* Because we're talking about forecasting and the mention of time-series forecasting, have you explored other conventional techniques like ARIMA commonly used in time-series forecasting?
* PeTEL seems like a schema of key-value pairs. Is there a fixed vocabulary or grammar syntax rules for this if calling it a language? (e.g. for POS tagging? What makes it a language vs a schema of key-values?
* Section 4.2 mentions “Given the lack of a pre-existing training dataset tailored to each unique schema/domain, fine-tuning pre-trained models is unattainable.” But the methodology presented in the paper involves fine-tuning. So isn't this contradictory?
* Were there any ablation studies performed on the effect of the 3 versions of t5 model data mixtures used? What is the relevance/necessity of adding those 3 mixtures (ie. plain T5, T5+1k examples, T5+10k examples)? And isn't there a bias introduced in the template structure by adding in that mixture for fine-tuning the T5?
* Algorithm 3 in appendix does not mention how attributes are used to create the utterance in the keyword-to-user utterance task. Is this done for each attribute (i.e. 1 utterance per attribute?) or a subset of attributes (ie. 1 utterance for a randomly selected subset of attributes?). This is a crucial detail that's missing.
* Algorithm 1 states the sorted list of attributes is shown to the user and then based on if the user agrees or not, proceed. But isn’t the whole point that the user does not know about data science (and so they don’t know about the target attribute)? So why have them evaluate?
* Section 6.1 is confusing. What does the count have to do here? Each inference is independent of the other, there’s no history being tracked, so what is meant by convergence? Is there some sort of majority voting done from multiple runs to get the final answer? If so, is this doesn’t seem to be explained anywhere.
* There are too many examples of ChatGPT’s output to qualitatively analyze the performance, but this isn’t of much relevance to the paper. Include more of the examples of the BERT Based models used. What do the generated/synthetic utterances look like? What does the CoNNL-like annotation dataset look like? What do the responses look like?
* Almost all of A.1 is in Section 4.2. Please dedupe.
* The main point in the paper is using this approach works for unseen datasets. In that case, shouldn’t the evaluations be done on datasets that are completely unseen during training? I.e. no fine-tuning done on them using custom-crafted heuristic datasets? Maybe use 2 of those datasets in training and validation, but keep one dataset completely untouched and only use it for evaluation.
* I'm a bit confused: Why to even extract the phrase? If the utterances are being generated based on a template that itself just adds in attributes, why can’t each example in the dataset be annotated with the attributes directly and predict that directly instead of needing the phrase? There aren't any implicit utterances in the dataset anyway.

**Strengths And Weaknesses:**

**Strengths**
* The paper tackles an interesting problem domain of democratizing datascience tools by leveraging advances in ML, LLMs and NLP.
* The paper prudently notes that the current generation of decoder-only models like ChatGPT might be quite performant in this domain, and identify this as an avenue for future work.

**Weaknesses**
* The paper claims to introduce the concept of forecast utterance, but it doesn’t seem well-explained or novel. It isn't different from a regular input to an NLP task. So there's no added relevance for introducing it.
* The paper claims a zero-shot learning approach. But it uses a heuristic created dataset to train the model. I'm not sure if this qualifies it as being Zero-shot which requires there are no examples used to train, and inference is run directly. And this wouldn't work for unseen datasets. It requires constructing those templates that seem specific to the datasets used. For e.g. in section 4.2, the template is about airlines, specific to that dataset. How can that be used for a different dataset? If custom hand-crafting of those templates was done for each dataset, then it cannot be used for any unseen dataset - and hence not zero shot.
* The claim about use of NER techniques is not proven clearly. The abstract states that 2 techniques are used. Appendix A.4 says 3 popular NER techniques are used. But none of their details or citations are mentioned. IIUC a BERT model is fine-tuned to output salient phrases from the utterance, and then selecting the attribute in that phrase. Appendix A.6 again talks about EE and QA tasks, but there are no examples of this in the Appendix.

---

> ### Author Response · Authors · 2024-01-04
> **Discussion and Clarification of Reviewer's concerns**
>
> Thank you for your insightful comments and suggestions. We have carefully considered each point and would like to address them as follows:
>
> 1. **Diversity of Data Science Needs**: Our vision focuses on accommodating non-expert machine learning (ML) users, such as small business owners or domain experts, who have unique datasets and wish to apply time series forecasting on them. These users might prefer to articulate their forecasting tasks in natural language, despite being content with a limited range of techniques, with the main objective of employing these techniques on their own datasets. This situation necessitates a conversational agent that is capable of understanding these specific datasets through natural language interaction, as users are often not well-versed in ML. The challenge is that creating a pre-existing, comprehensive training dataset for such conversational agents is impractical due to the diverse and unique nature of user datasets. Hence, our approach is designed to meet this unique user-need profile, ensuring that non-expert users can effectively use ML applications tailored to their specific business needs.
>
> 2. **ARIMA**: Our paper aims to enable users to articulate forecasting tasks through conversation, rather than focusing on executing a specific forecasting method like ARIMA. The conversational agent’s role is to guide users in defining their forecasting needs. This differs from the perceived task of directly running forecasting techniques. Our goal is to bridge communication between users and forecasting tools, allowing for a seamless formulation of tasks that can be executed by any method, including but not restricted to ARIMA. Given a particular method, the model will try to infer the hyperparameters and parameters of that model to its best capacity. For example, given a dataset with 12 attributes, just saying use ARIMA for forecasting is not enough; we need to mention what is the target variable, what is the lead time, what is the prediction window and so forth before an ARIMA model can be actually trained. This approach highlights the innovative aspect of our research, centered on the conversational interface to understand forecasting “needs” rather than the forecasting “methods” themselves.
>
> 3. **Nature of PeTEL**: PeTEL is a simple a key-value pair data structure. We understand that calling it a “language” can be confusing and we are more than willing to revise the terminology.
>
> 4. **Fine-Tuning Methodology**: We agree with your comment that it can appear a to be confusing. What we wanted mean is that fine-tuning with real user utterance data is not possible, and we have to fine-tune with synthetic examples. We call it zero-shot becuase no new real data points are required. But as you mentioned, you are happy to renaming from “zero-shot” to “unsupervised” if you would prefer that.
>
> 5. **T5 Model Data Mixtures**: In Section 4.2, we discuss the rationale behind using three variations of T5 models. These variations are essential to our methodology, enabling us to analyze the effectiveness of different training data scales.
>
> 6. **Keyword-to-User Utterance Task in Algorithm 3**: We agree with your comment that this detail is currently missing. To elaborate, in Algorithm 3's Keyword-to-User Utterance Task, as discussed in Section 4.2, we perform the commonGen Task to generate user utterances from keywords. To clarify this process, we will include a detailed description of the commonGen task in the revised version.
>
> 7. **User Involvement in Attribute Selection**: The involvement of users, especially those without data science knowledge, is a cornerstone of our approach in formulating a forecasting task. Note that, although the users don’t know data science, they are the experts in their own domains and have a better understanding of the dataset themselves than a professional data scientist. By actively conversing with the user, we not only ensure that the forecasting task is tailored to their specific needs and preferences, but also foster a sense of involvement and ownership. This approach allows users to create a forecasting task that truly resonates with their individual requirements, rather than merely accepting a recommendation. This inclusive strategy is vital for aligning the forecasting task with the user’s unique context and objectives.

---

> > ### Author Response · Authors · 2024-01-04
> > **Discussion and Clarification of Reviewer's concerns Continued**
> >
> > 8. **Convergence in Section 6.1**: Convergence in this context refers to the number of iterations required for the agent to accurately identify the target slot, indicating the model's efficiency in understanding and responding to user needs, particularly with implicit slot values. In our experiments, extractive models are utilized to extract salient phrases for ranking available attributes. The count of iterations needed for accurate identification of the target slot value is a key measure of convergence. This resembles the idea of “mean reciprocal rank”. The data shows that if the extracted phrase is not relatable, convergence is slower, a trend especially noticeable with implicit slot values in utterances.
> >
> > 9. **Inclusion of ChatGPT and BERT-Based Model Examples**: In the revised version, we will include examples of the BERT-based models used to enhance the overall understanding of the study.
> >
> > 10. **Duplication in Appendix A.1**: We acknowledge the redundancy and will revise Appendix A.1 to eliminate duplication with Section 4.2.
> >
> > 11. **Evaluations on Unseen Datasets**: Our methodology inherently involves creating a synthetic dataset after receiving a user's dataset, which is integral to our fine-tuning process. What we offer is a “real time lightweight fine-tuning" rather than intensive fine-tuning.  Therefore, evaluating on datasets completely unseen during training would deviate from our proposed methodology's core premise.
> >
> > 12. **Necessity of Extracting Phrases**: We think you have mixed up two different things here. The distinction between the synthetic dataset for training and the handcrafted evaluation dataset is crucial. The extraction of phrases is part of our evaluation phase to assess the system's performance on real-world, implicit utterances, which is separate from our synthetic dataset generation that is done for fine-tuning.
> >
> > We hope these responses adequately address your concerns and clarify our methodology. We are grateful for the opportunity to enhance our paper with your feedback.

---

### Decision · Action_Editor_gkgA · 2024-02-06

**Recommendation:** Accept with minor revision

**Comment:**

The authors should make efforts to address the reviewers' comments in the revision. Particularly, Reviewer rYGP pointed out three weaknesses. It is important for the authors to clarify the nature of the techniques. Are they really zero-shot learning? Is hand-crafted dataset needed for each specific domain? If so, how difficult is it to construct them --- presenting them differently can improve clarity of the paper?

**Audience:**

The work should be of interest of the audience in TMLR, particularly those working on bringing LLMs to specific domains.

**Claims And Evidence:**

All the reviewers think the paper addresses an important problem of facilitating data science tasks by leveraging the advances of LLMs. The authors introduced a set of new techniques to address the specific domain of time series prediction. The work seems well evaluated Although the presentation of the work has a number of issues, these can be fixed by improving writing and clarify for the final version.